# in situ observation of reversible phase transitions in Gd-doped ceria driven by electron beam irradiation

Ke Ran [1,2,3] ✉, Fanlin Zeng[4], Lei Jin[2], Stefan Baumann [4], Wilhelm A. Meulenberg [4,5] & Joachim Mayer[1,2]

Ceria-based oxides are widely utilized in diverse energy-related applications, with attractive functionalities arising from a defective structure due to the formation of mobile oxygen vacancies ($V_O^{..}$). Notwithstanding its significance, behaviors of the defective structure and $V_O^{..}$ in response to external stimuli remain incompletely explored. Taking the Gd-doped ceria ($Ce_{0.88}Gd_{0.12}O_{2-\delta}$) as a model system and leveraging state-of-the-art transmission electron microscopy techniques, reversible phase transitions associated with massive $V_O^{..}$ rearrangement are stimulated and visualized in situ with sub-Å resolution. Electron dose rate is identified as a pivotal factor in modulating the phase transition, and both the $V_O^{..}$ concentration and the orientation of the newly formed phase can be altered via electron beam. Our results provide indispensable insights for understanding and refining the microscopic pathways of phase transition as well as defect engineering, and could be applied to other similar functional oxides.

Owing to the flexile valence switching between $Ce^{4+}$ and $Ce^{3+}$ cations and the facile formation of oxygen vacancies ($V_O^{..}$)[1–4], ceria has been recognized as one of the best candidates for catalysts[5–7] and solid electrolytes[8,9]. Ceria-based catalysts with stabilized and adequate active sites are attracting continuous attention[10,11], and important advances have also been made towards the production of sustainable and clean energy[12,13]. Besides, memristors based on ceria are a critical component for next-generation nanoelectronics. The conductive species can be effectively controlled by external field to readily achieve an ON/OFF switching, which have been extensively explored for possible large-scale integrated circuits[14–17]. Regarding all these functional oxide materials and their associated devices, the formation and migration of $V_O^{..}$ under external stimuli constitute the fundamental processes, which are often linked with modifications of the ceria structure and subsequent phase transitions. Typically, ceria crystallizes in a fluorite-type (F-type) cubic structure (space group $Fm\bar{3}m$, a ≈ 5.42 Å), where the metal site (M) sitting at (0,0,0) is coordinated to eight O at (1/4,1/4,1/4)[18].

Transition from F-type to the so-called C-type (space group $Ia\bar{3}$) takes place when enough $V_O^{..}$ and $M'_M$ are introduced to the system[19–22]. As a result, the cell parameter is doubled, the M site is six-coordinated to O, and the crystallographic positions are split[23,24]. The C-type structure is described as $M_{1-x}^{4+}M_x^{3+}O_{2-\delta}^{2-}$, where $\delta = x/2$ is the concentration of $V_O^{..}$ and $M_x^{3+}$ is trivalent ions like $Gd^{3+}$ and/or reduced $Ce^{3+}$.

Benefiting from the cutting-edge transmission electron microscopy (TEM) techniques, significant insights into the transition process have been acquired, providing outstanding spatial, chemical and temporal resolution[10,13,25,26]. By in situ environmental TEM (ETEM), the redox process in ceria nanoparticles was studied at elevated temperature and in $H_2/O_2/CO_2$ environment[27–30]. At ambient temperature, in situ electrical probe TEM was also able to drive reversible resistance switching by applying electrical field[16,31]. Nevertheless, considering the significance of phase transitions in the context of technological applications, the reported findings still lack elaborate investigations or adequate spatial/elementary resolution, primarily due to the

[1]Central Facility for Electron Microscopy GFE, RWTH Aachen University, Aachen, Germany. [2]Ernst Ruska-Centre for Microscopy and Spectroscopy with Electrons ER-C, Forschungszentrum Jülich GmbH, Jülich, Germany. [3]Advanced Microelectronic Center Aachen, AMO GmbH, Aachen, Germany. [4]Institute of Energy and Climate Research IEK-1, Forschungszentrum Jülich GmbH, Jülich, Germany. [5]Faculty of Science and Technology, Inorganic Membranes, University of Twente, Enschede, AE, The Netherlands. ✉e-mail: ran@gfe.rwth-aachen.de

challenges associated with real-time visualization of dynamic O atoms. Questions pertaining to transition details, including the selection of external stimuli, the behaviors of $V_{\ddot{O}}$, the feasibility of fine tuning, and the underlying mechanism remain unsolved.

Herein, the Gd-doped ceria (CGO)[32,33] with high oxygen ion conductivity and giant electrostriction under external electric field[34,35] is chosen for our study (Ce$_{0.88}$Gd$_{0.12}$O$_{2-\delta}$, in Supplementary Note 1 and Fig. 1). Utilizing the incident electron beam (e-beam) as an external stimulus, the phase transition of CGO is in situ probed down to sub-Å scale by TEM[36–39]. Varying electron dose rate (EDR), the transitions can be accelerated, retarded, hold, and reversed. Negative spherical aberration imaging (NCSI)[40–42] and integrated differential phase contrast (iDPC)[43] imaging are employed, enabling high contrast for both light oxygen and heavy metal atoms, as well as facilitating the measurement of atomic positions with ultrahigh precision. Quantifying lattice distortions also allows a direct estimation of the local $V_{\ddot{O}}$ concentration. Together with the proposed mechanism, our findings are crucial for the engineering and optimization of numerous energy-related applications relying on ceria and analogous oxides.

## Results and Discussion
### Reversible transition between F- and C-type CGO
Figure 1a sketches the experimental design for in situ transition study. In TEM mode, e-beam with high EDR is able to stimulate the F-to-C transition, while the C-to-F reverse transition is assisted with a low EDR. Both F- and C-type models are viewed along < 001 > in Fig. 1a, and having 30% of the M sites occupied by trivalent ions (CGO30). As mentioned earlier, comparing with F-type, the cell parameter of C-type is doubled (the solid squares in Fig. 1a). Furthermore, part of the M positions in the C-type (the pairs of lines in Fig. 1c) and all its O positions are splitting evidently. Based on the CGO30 models, high-resolution TEM (HRTEM) images are simulated[44], in Figs. 1b–c. As plotted on the left and right side in Fig. 1b, identical peak intensities from the M positions and constant $d_{O\_v}$ (the distance between two neighboring O along the vertical direction) are revealed from the F-type. In contrast, the split M positions in Fig. 1c leads to a higher intensity peak than the unsplit ones, and the measured $d_{O\_v}$ varies in a breathing-like manner.

Similar features are observed by experiment as well. Figure 2a–c shows the iDPC results from the F-type along <001 >. As a F-to-C transition is usually unavoidable under the HRTEM imaging condition, iDPC technique is therefore employed for safely imaging the F-type structure (Supplementary Note 2 and Fig. 2). In Fig. 2a, both the M and O sites can be clearly resolved, showing a good agreement with the embedded F-type model and the simulated iDPC image[44]. Moreover, on the left side of Fig. 2a, the laterally averaged intensity profile shows constant M peaks, similar as in Fig. 1b. Each M and O position in Fig. 2a are then determined using two-dimensional Gaussian fitting[45]. In Figs. 2b, c, the mapped distances between neighboring M and O positions ($d_M$ and $d_O$) are overlaid on the iDPC image. On the right side of Fig. 2c is the laterally averaged $d_{M\_v}$ and $d_{O\_v}$ as defined in Fig. 2a, where both values are almost constant. Figure 2d–f are the corresponding HRTEM results from the C-type. Consistent with the simulation in Fig. 1c, alternating M peaks are noticed in the intensity profile on the left side of Fig. 2d. The mapped $d_{M\_v}$ in Fig. 2e are still constant, while the $d_{O\_v}$ in Fig. 2f are regularly oscillating.

The additional ordering in the C-type structure also causes extra spots in the diffraction pattern, as those in Fig. 2g, which were recorded during a transition cycle. Marked by the circles, extra spots only show up for the C-type, and are indexed as {010} following the F-type. Figures 2h and 2i are the FFT patterns from Figs. 2a, d, respectively. In agreement with the diffraction patterns, only C-type is associated with extra spots. It's also noticed in Fig. 2i that there are only two {010} spots along the vertical direction, while in Fig. 2g the {010} spots are located along both the vertical and lateral direction. This difference can be explained by the two perpendicular orientations of the C-type, which will be discussed later.

### Modulating the phase transition
As shown in Figs. 2g, i, the extra spots in either diffraction or FFT patterns enable a direct detection of the C-type during the phase transition. Given that the transition is a rather fast process, yet acquiring diffraction patterns is relatively time consuming, the FFT pattern based on TEM image acquired with short exposure time but sufficient spatial resolution is thus employed to in situ study the phase transition. Two cases are shown in Figs. 3a, b (F-to-C) and Figs. 3c, d

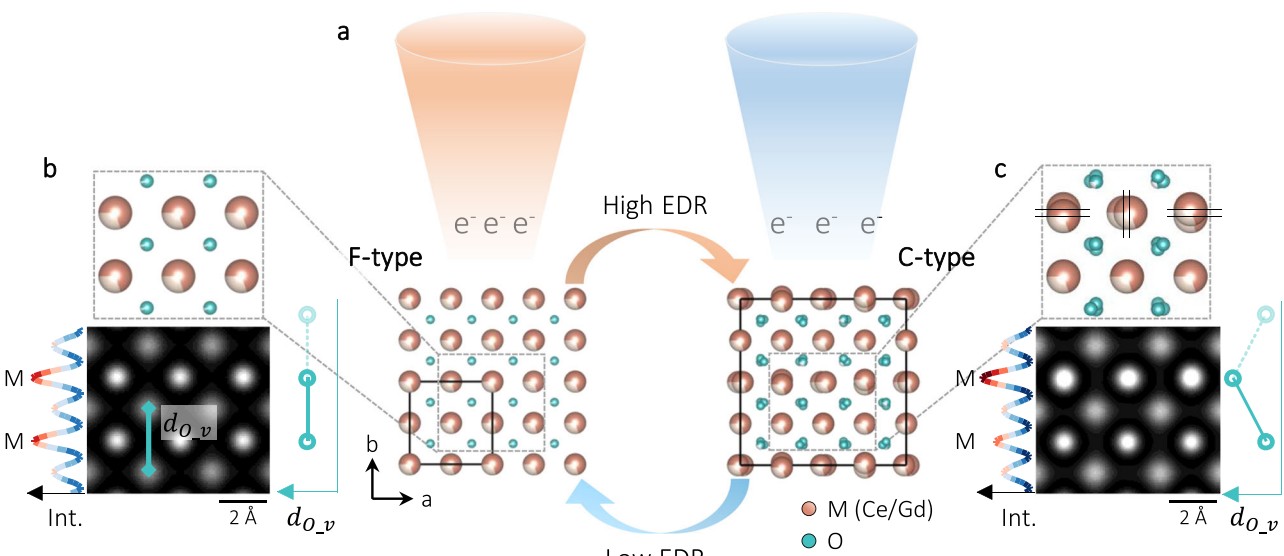

**Fig. 1 | The reversible transition between F- and C-type. a** Schematic of the reversible transition: in TEM mode, e-beam with high and low EDR is used to stimulate the F-to-C and C-to-F transition, respectively. F- and C-type CGO30 models are viewed along < 001 > at the bottom of (**a**). Single unit cells of each type are outlined by the solid squares. **b**–**c** Enlarged models from the dashed rectangles in **a**.

The three sets of parallel lines in (**c**) indicate split metal positions. Below the models are the simulated HRTEM images. Laterally averaged intensity profile and the estimated $d_{O\_v}$ (virtually extended as the dotted lines) are plotted on the left and right side of the simulated images.

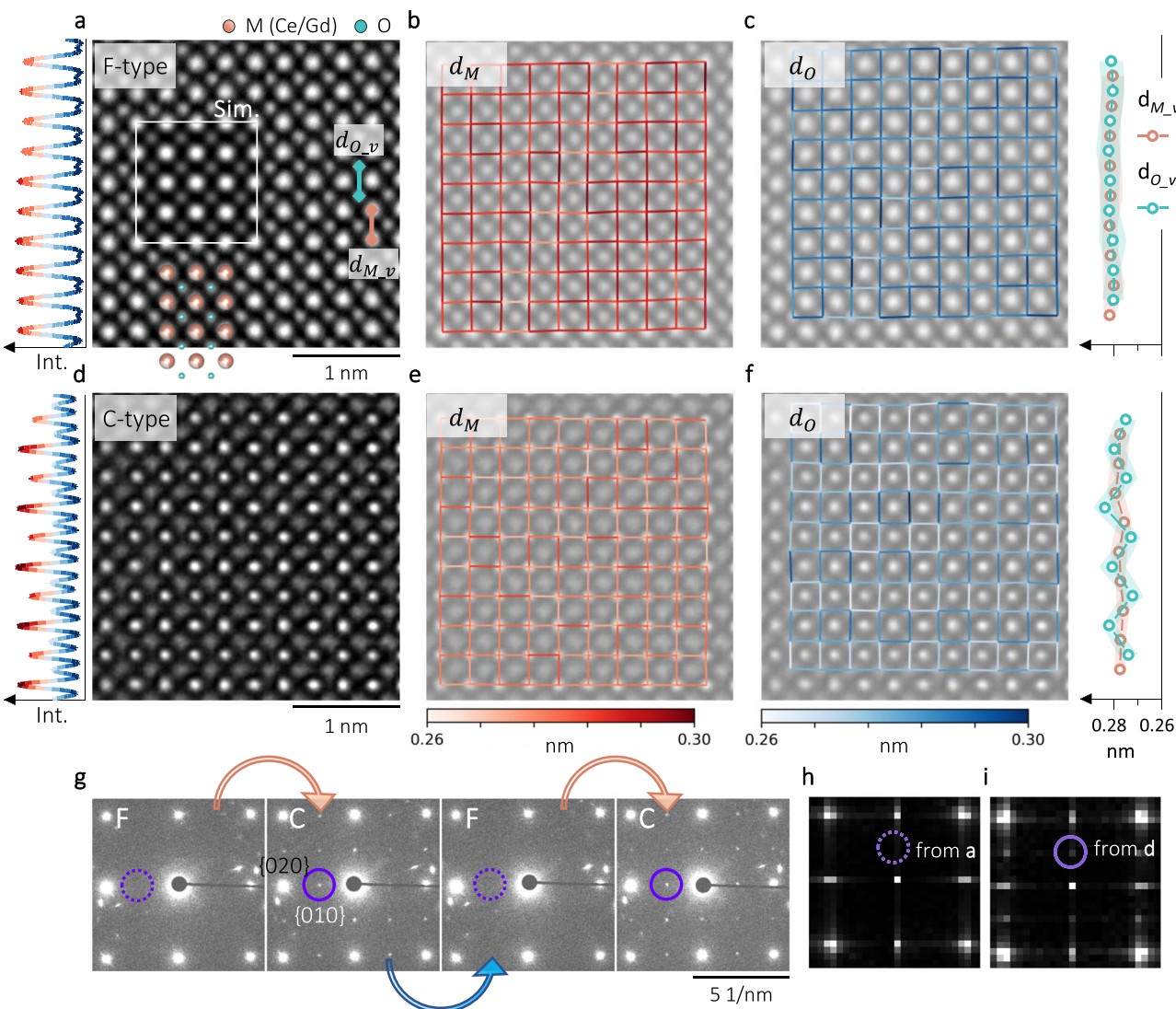

**Fig. 2 | Experimental visualization of both F- and C-type. a** iDPC image of the F-type along < 001 > , together with a F-type model and the simulated iDPC image. On the left is the laterally averaged intensity profile. **b–c** The mapped distances between neighboring M and O positions ($d_M$ and $d_O$) based on (**a**). On the right side of (**c**) are the laterally averaged $d_{M\_v}$ and $d_{O\_v}$. **d–f** The corresponding HRTEM results from the C-type. **g** Diffraction patterns from a transition cycle. Circles are placed at the same position for all the patterns. **h–i** FFT patterns from (**a** and **d**).

(C-to-F). In Figs. 3a, e-beam with high EDR (2656 e · Å$^{-2}$ · s$^{-1}$, and noted as 0.69 relative to the highest EDR 3825 e · Å$^{-2}$ · s$^{-1}$ listed in Fig. 3, to which all the EDRs are normalized, Supplementary Table 1) is used for inducing the phase transition and imaging. A series of TEM images were recorded (Supplementary Fig. 3), and the corresponding FFT patterns at different time are listed in Fig. 3b. Weak {010} spots can already be recognized at ~10 s. As the irradiation continues, the {010} spots get continuously stronger. A much lower EDR (noted as 0.06) is used in Fig. 3c for the C-to-F reverse transition. Due to both the weak beam and low magnification, the FFT in Fig. 3d is rather noisy, when comparing with Fig. 3b. Nevertheless, the {010} spots are still visible at 0 s and start to fade away after ~26 s under continuous low EDR illumination, as revealed by the dashed circles. At 78 s, the extra spots are no longer detectable suggesting the recovery of the F-type structure.

Under the thin sample condition, the intensity of {010} spots are directly associated with the amount of C-type within the irradiated region. The intensity ratio between the four {010} and the four {020} spots, denoted $r$, can thus be used to evaluate the transition process. Several image time series were acquired with various EDRs. Figure 3e plots the $r$ as a function of time for all of the F-to-C transitions. Each

curve in Fig. 3e is divided into several regions for linear fitting (the solid lines), and the fitted slopes are listed in Fig. 3h, shedding light on the efficiency of each transition. Overall, the higher the EDR, the faster the F-to-C transition. Continuous irradiation can speed up the transition, as the slopes are noticed increasing with time for each EDR. One exception is EDR 1.0, where the fitted slope drops from 0.013 to 0.003 after ~14 s irradiation. Moreover, the intensity ratio $r_A$ and $r_B$ between the two {010} and the two corresponding {020} spots along the indicated A and B directions in Fig. 3b are also plotted in Fig. 3f, to study the anisotropy of the phase transition. Both $r_A$ and $r_B$ are growing at a similar pace at the beginning for all the transitions. After a certain point depending on the EDR, $r_A$ prevails, suggesting that the C-type structure favors orienting with its b axis (as defined in Fig. 1a) along A direction in the present case (Supplementary Note 3 and Fig. 4). Besides, the apparent deviations between $r_A$ and $r_B$ are found to be closely correlated with the change of slopes in Fig. 3e. For instance, an evident split of $r_A$ and $r_B$ is observed at ~14 s for EDR 1.0 in Fig. 3f, which coincides with the abrupt drop of the slope from 0.013 to 0.003 as shown in Fig. 3h. In contrast, the C-to-F reverse transitions are slower and show less dependence on the EDRs, as shown in Fig. 3g. The relatively large values of $r$ are mainly

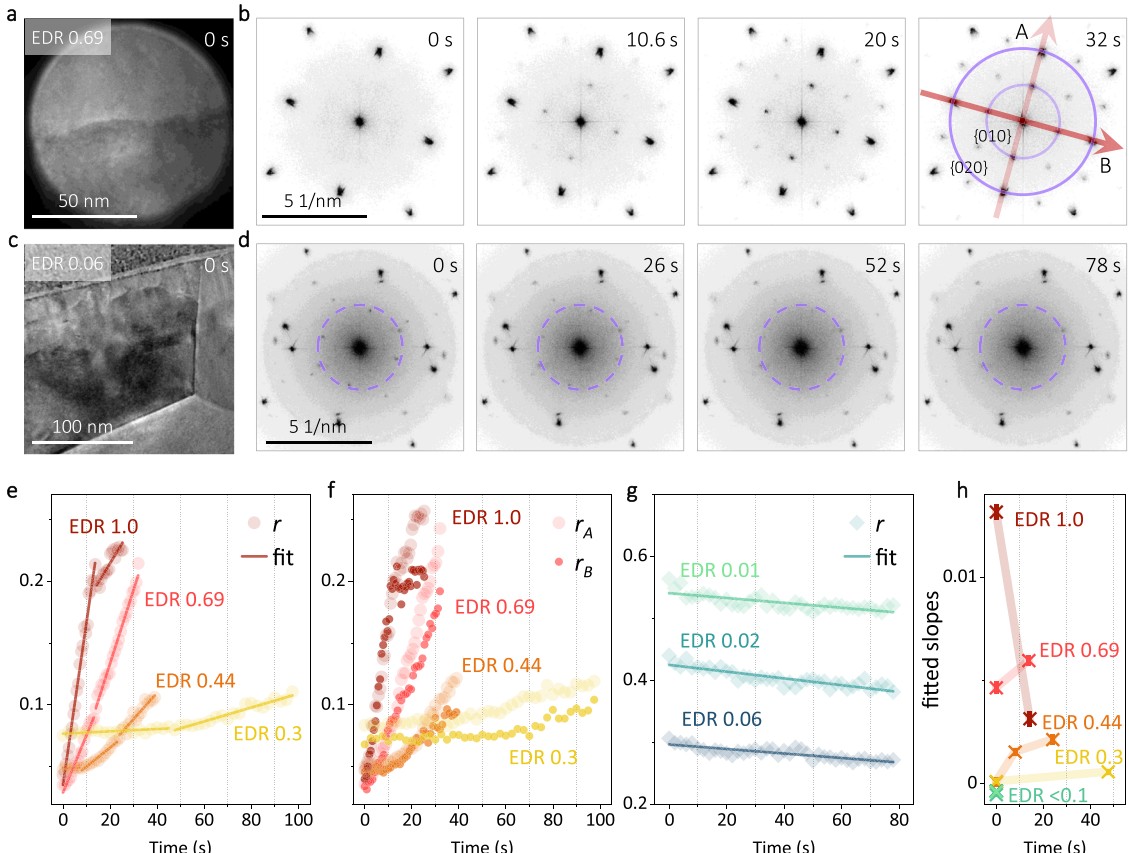

**Fig. 3 | Transitions with different EDRs. a, b** One F-to-C transition with EDR 0.69 (relative value, where the highest EDR 3825 e·Å$^{-2}$·s$^{1}$ listed in Fig. 3 is noted as EDR 1.0). A series of 40 TEM images were recorded with 0.04 s exposure time and 0.8 s interval. The image at the outset and the FFT patterns (absolute value) during the transition are shown. **c–d** One C-to-F transition with EDR 0.06. A series of 40 TEM images were recorded with 0.04 s exposure time and 2 s interval. The image at the outset and the FFT patterns (absolute value) during the transition are shown. **e–g** $r$ (intensity ratio between the four {010} and the four {020} spots), $r_A$, and $r_B$ (intensity ratio between the two {010} and the two {020} spots along A and B) as a function of time from each transition. Linear fitting is applied to $r$, and the estimated slopes are listed in (**h**).

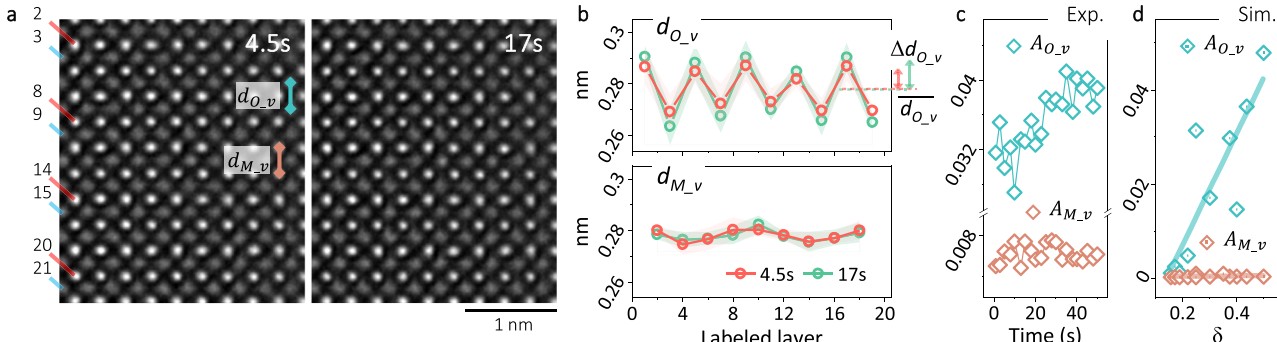

**Fig. 4 | Pushing the $V_O^{..}$ ordering.** A series of 50 HRTEM images was recorded with 0.5 s interval. Two of them are shown in (**a**). **b** The determined $d_{O\_v}$ and $d_{M\_v}$ from each labeled layer, as defined in (**a**). **c** $A_{O\_v}$ and $A_{M\_v}$ as a function of time, based on the HRTEM image time series. **d** $A_{O\_v}$ and $A_{M\_v}$ estimated from the simulated HRTEM images. The image simulations are based on C-type models with varying $\delta$ from other studies[21,23,46,47]. Solid lines suggest linear fits.

resulting from the high noise level in the FFT patterns associated with low EDRs. The fitted slopes are plotted in Fig. 3h as well, where the values are almost identical for all the three transitions.

## Modifying the C-type

After its formation, further modifying the C-type structure is also feasible. As already shown in Fig. 2f, the breathing-like oscillation of $d_{O\_v}$ is a robust signature of the C-type, and the amplitude of the oscillation is reported to be dependent on the $V_O^{..}$

concentration, $\delta$ [21,23,46,47]. Thus, via monitoring the $d_{O\_v}$ oscillation, $\delta$ variation under the e-beam can be revealed.

Figure 4a shows two images from an HRTEM image time series. The defined $d_{O\_v}$ and $d_{M\_v}$ are averaged within each labeled layer and plotted in Fig. 4b for both images. The $d_{M\_v}$ is rather constant among the labeled layers and also close between the two images. The $d_{O\_v}$ is oscillating as expected. However, the oscillation at 17 s is obviously broader than that at 4.5 s. In order to track the oscillation in a quantitative way, $A_{O\_v} = \triangle d_{O\_v} / \overline{d_{O\_v}}$, is calculated from each image in the

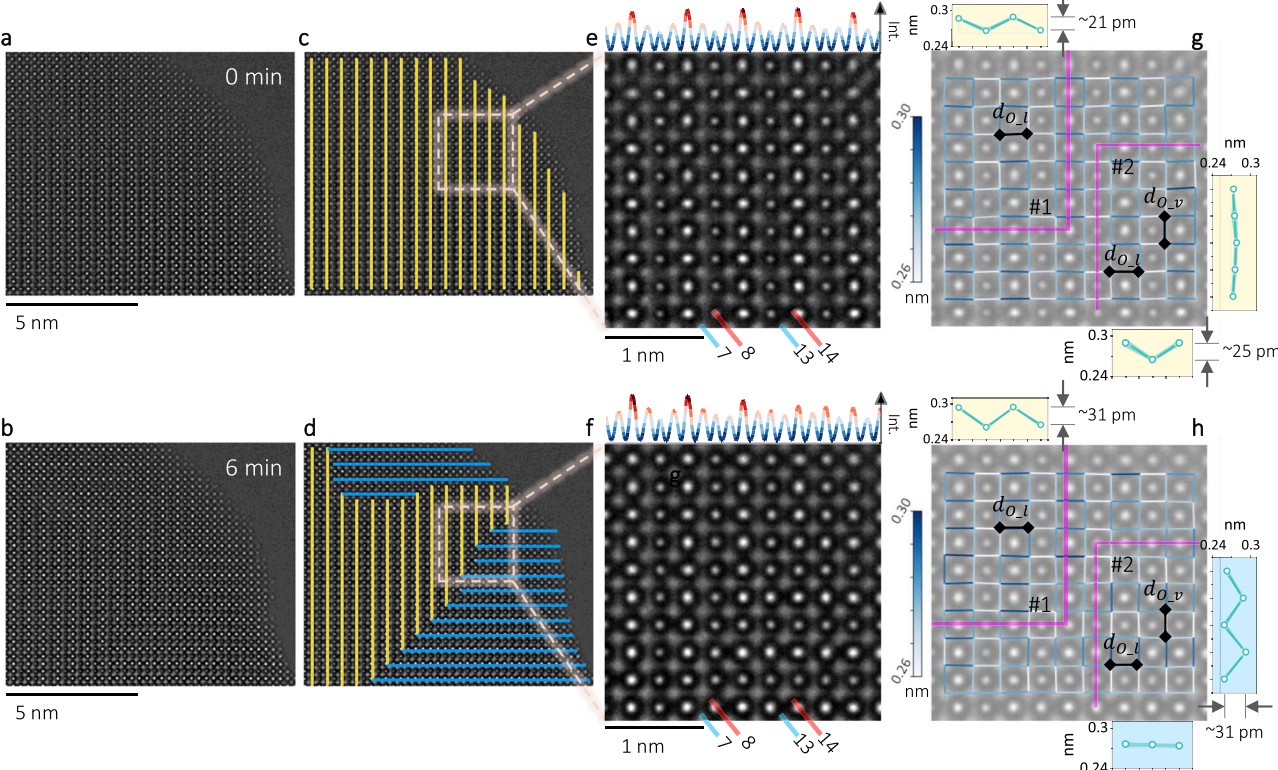

**Fig. 5 | Rotating the C-type structure. a**, **b** HRTEM images of the C-type structure before and after e-beam irradiation. **c**, **d** The same as in (**a**, **b**) with the brighter metal layers outlined by yellow/vertical and blue/parallel lines. **e**, **f** The enlarged images from the squared regions in (**c**, **d**). At the top is the vertically averaged intensity profiles, and atomic layers are labeled at the bottom. **g**–**h** The mapped distances between neighboring O positions, based on (**e**, **f**). Two regions (#1 and #2) are defined by the lines in magenta. From region #1, vertically averaged $d_{O\_l}$ are plotted at the top. From region #2, vertically averaged $d_{O\_l}$ and laterally averaged $d_{O\_v}$ are plotted at the bottom and on the right, respectively.

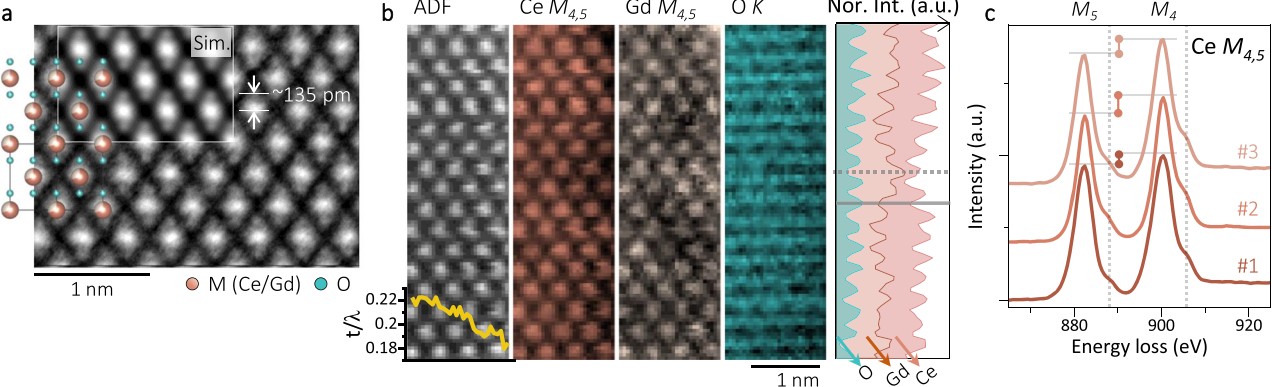

**Fig. 6 | Chemical structure of CGO. a** iDPC image of the CGO along <110> direction. A F-type model and the simulated iDPC image are overlaid. **b** EELS SI results: simultaneously acquired ADF image and elemental maps plotting the intensity from the Ce $M_{4,5}$, Gd $M_{4,5}$ and O $K$ edge. At the lower-left corner is the vertically averaged t/λ profile, and on the right is the laterally averaged intensity profiles from the O, Gd and Ce map. **c** Fine structure of the Ce $M_{4,5}$ edge from three different regions (#1 to #3).

series, where $\triangle d_{O\_v}$ and $\bar{d}_{O\_v}$ are the deviation and mean value of all the measured $d_{O\_v}$ from a single image. On the right of Fig. 4b, $\triangle d_{O\_v}$ (the double-headed arrows) and $\bar{d}_{O\_v}$ (the dotted lines) determined from Fig. 4a are indicated. Similarly, $A_{M\_v}$ for the M positions are determined as well. Both $A_{O\_v}$ and $A_{M\_v}$ are plotted as a function of time in Fig. 4c. Evidently, $A_{O\_v}$ is increasing with prolonged irradiation, while $A_{M\_v}$ shows ignorable changes with time.

The results in Fig. 4c is further compared with the C-type models with varying $\delta$ from other studies[21,23,46,47]. In order to

enable a direct comparison, HRTEM images are first simulated based on these models (Supplementary Note 4 and Fig. 5). From these simulated images, $A_{O\_v}$ and $A_{M\_v}$ are estimated as well, and plotted as a function of $\delta$ in Fig. 4d. Similar to Fig. 4c, $A_{M\_v}$ keeps constant within the whole range in Fig. 4d, while $A_{O\_v}$ increases almost monotonously as a function of $\delta$. Correlating the values of $A_{O\_v}$ between Figs. 4c and 4d allows a direct determination of the $\delta$ within the irradiated region. Explicitly, the local $\delta$ can be driven up to 0.44 by e-beam in the present case.

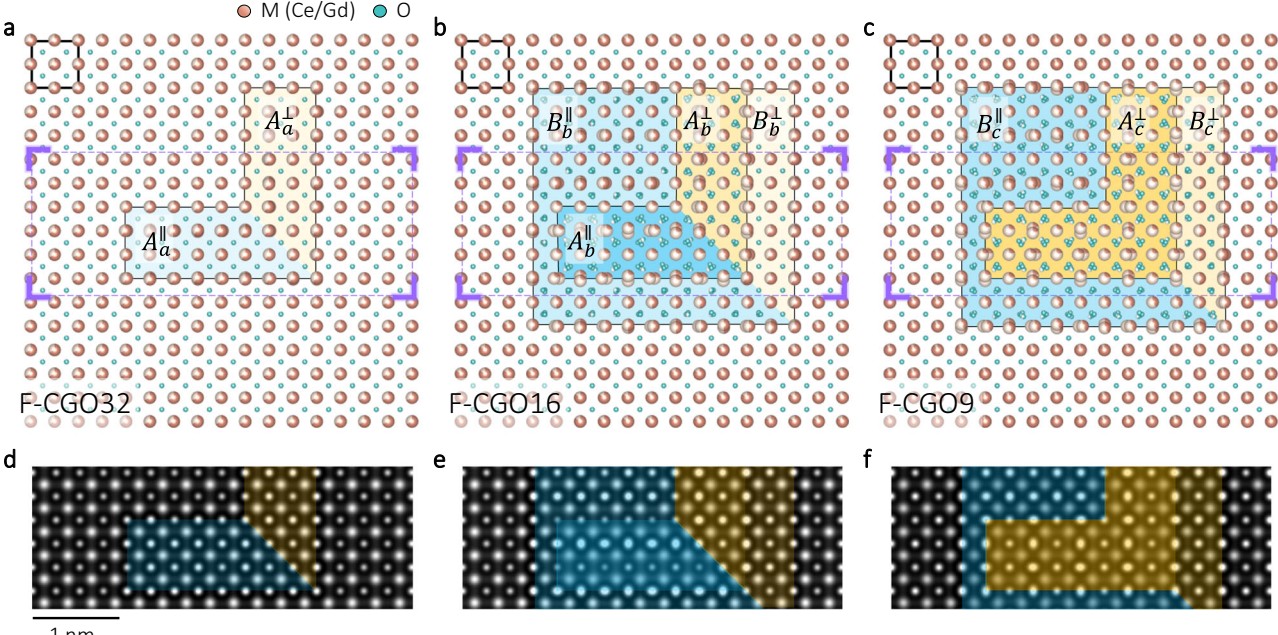

**Fig. 7 | Proposed F-to-C transition.** The C- and F-type are separated by the colored backgrounds, and one unit-cell of the F-type is outlined at the top-left corner. **a** Early stage of the transition: two small regions ($A_a^{\parallel}$ and $A_a^{\perp}$) take the C-type structure (C-CGO40, 40% of the metal sites are occupied by M³⁺) with a relative rotation of 90°. **b** Two more regions transfer to the C-type: $B_b^{\parallel}$ (C-CGO50) and $B_b^{\perp}$ (C-CGO40). Meanwhile, the $A$ regions evolve to $A_b^{\parallel}$ (C-CGO80) and $A_b^{\perp}$ (C-CGO60).

**c** Further irradiation raises the $V_O^{\bullet\bullet}$ concentration in the $B$ and $A^{\perp}$ region, where $B_c^{\parallel}$, $B_c^{\perp}$ and $A_c^{\perp}$ become C-CGO60, C-CGO50, and C-CGO80. Besides, the $A_b^{\parallel}$ region experiences a 90° rotation and are combined with $A^{\perp}$. Composition of the F-type substrate changes accordingly, as indicated at the lower-left corners. **d**–**f** Simulated HRTEM images corresponding to the regions defined by the rectangles in (**a**–**c**).

In Fig. 3f, the C-type structure is observed orienting along two perpendicular directions, but not equally developed as a function of time (see divergence of $r_A$ and $r_B$). The competition between the two orientations is also revealed by HRTEM image shown in Figs. 5a, b, where a massive rotation of 90° is manifest. As already shown in Fig. 2d, along the vertical direction, intensity of the neighboring M layers is alternating. Thus, the brighter M layers in Figs. 5a, b are marked by the yellow/vertical and blue/lateral lines in Figs. 5c, d, to highlight the two orientations. While the yellow lines are covering the whole image in Fig. 5c, up to half of Fig. 5d is occupied by the blue lines. The squared regions in Figs. 5c, d are further enlarged in Figs. 5e, f, with the intensity profiles plotted at the top. Consistent with the outlines in Fig. 5c, the M peaks are regularly oscillating in Fig. 5e. In contrast, such an oscillation breaks abruptly in Fig. 5f starting from Layer 12. Besides, distances between neighboring O positions, $d_O$, are mapped in Fig. 5g-h. Two regions, #1 and #2, are defined by the lines in magenta. Within region #1, the vertically averaged $d_{O\_l}$ (the distance between neighboring O positions along the lateral direction) as plotted at the top suggest similar oscillations in both images, only the gap is increasing from ~21 pm in Fig. 5g to ~31 pm in Fig. 5h. For region #2, the vertically averaged $d_{O\_l}$ at the bottom shows an oscillation with a gap of ~25 pm in Fig. 5g but a rather flat line in Fig. 5h. On the contrary, a flat line is obtained on the right of Fig. 5g based on the laterally averaged $d_{O\_v}$, while an oscillation with a gap of ~31 pm shows up in Fig. 5h.

### Chemical structure of the CGO

Figure 6a shows the iDPC image of the F-type CGO along <110> direction, where the M and its nearest O positions distanced by ~135 pm can be clearly resolved. The experimental image is also in a good agreement with the overlaid F-type model and the simulated iDPC image. Electron energy loss spectroscopy spectrum imaging (EELS SI) with atomic resolution was recorded, in Fig. 6b, suggesting a successful Gd substitution of Ce and a uniform distribution of O. Due to the reducing sample thickness from left to right, as shown by the t/λ

profile at the lower-left corner in Fig. 6b (t: sample thickness and λ: electron inelastic mean free path), dropping intensity is noticed in both the ADF image and elemental maps. The laterally averaged intensity profile of Ce, Gd, and O are plotted on the right side of Fig. 6b, where Ce and Gd peaks show up simultaneously (the dotted line) while the O peak are between the Gd/Ce peaks (the solid line). Figure 6c further plots the fine structure of Ce $M_{4,5}$ edge from three different locations (#1-#3, Supplementary Fig. 6). Among the three curves, different features are noticed, including the slight shift of the shoulders (the dashed lines), and the varying height difference between the $M_5$ and $M_4$ edges (the bars with round ends). Based on the $M_5/M_4$ intensity ratio in second derivative spectra, the Ce³⁺ ratios (Ce³⁺/(Ce³⁺+Ce⁴⁺)) can be then determined as ~34%, ~2%, and ~18% for each curve respectively[32,48,49].

### The migration of $V_O$ and $Ce'_M$

The release and acquisition of O atoms in ceria are often considered essential for the reversible transition between F- and C-type, which is also paired with the ordering of $V_O^{\bullet\bullet}$ and reduction of Ce[50]. To release the O atoms from ceria nanoparticles, heating treatment (>900 K), H₂ environment, or very low O partial pressure (~10⁻²⁶ Pa) are required during in situ ETEM[27,28,30,51]. Our experiment took place at room temperature under $7.5 \times 10^{-6}$ Pa, and any heating effect by e-beam is rather ignorable (Supplementary Note 5 and Supplementary Table 1). Thus, the critical requirements by in situ ETEM can hardly be satisfied in our case. Intense e-beam could also remove the O atoms from the sample surface and generate $V_O^{\bullet\bullet}$s[20,52]. Usually, a threshold energy about tens of eV is required[53]. Here, the phase transitions were observed with different accelerating voltages: 60, 200, and 300 kV (Supplementary Fig. 7), and the maximum energy transfer during an elastic scattering process is ~8.7, ~32.8, and ~53.2 eV, respectively. Thus, in most cases, the transferred energy would be insufficient to knock the O out. Carefully examining the HRTEM image time series acquired at 300 kV also excludes any significant loss of atoms during imaging

**Table 1 | Compositions of the labeled regions in Figs. 7a–c**

|   | $A^{\parallel}$ | $A^{\perp}$ | $B^{\parallel}$ | $B^{\perp}$ | substrate | F-type | C-type |
|---|---|---|---|---|---|---|---|
| a | C-CGO40 | C-CGO40 | F-CGO32 | F-CGO32 | F-CGO30 | 85% | 15% |
| b | C-CGO80 | C-CGO60 | C-CGO50 | C-CGO40 | F-CGO16 | 57% | 43% |
| c | C-CGO80.**r** | C-CGO80 | C-CGO60 | C-CGO50 | F-CGO9 | 57% | 43% |

(Supplementary Fig. 8)[54]. Additionally, a redox process can be driven by electric field. By in situ biasing TEM[16,31,37], reversible migration of $V_O^{\cdot\cdot}$ and the associated phase transition have been reproducibly achieved. Similarly, an electric field can be built up by e-beam irradiation[55,56] and triggers the phase transition[38,57]. However, the built-up electric field under the TEM illumination will attract $O^{2-}$ to the irradiated region, opposite to the rising $V_O^{\cdot\cdot}$ concentration as observed. Thus, such a field should only have limited contribution to our observation. Moreover, the phase transition is only detected in TEM mode, but not in STEM mode. Considering the different illumination geometry, a parallel beam (>100 s nm) in TEM mode and a focused probe (<0.1 nm) in STEM mode, it is reasonable to interpret the phase transition as a collective rearrangement of a large number of $V_O^{\cdot\cdot}$s stimulated by the incident electrons. Obviously, the ultrafine STEM probe is unlikely to interact with many $V_O^{\cdot\cdot}$s at the same time, and cause detectable structural change.

The exact mechanism which accounts for the phase transition is still to be defined. Qualitatively, the phase transition consists of two competing processes: accumulating and ordering of the $V_O^{\cdot\cdot}$ towards the irradiated region (P1, the forward F-to-C transition), and dissipating and disordering of the $V_O^{\cdot\cdot}$ from the irradiated region (P2, the reversed F-to-C transition). Both P1 and P2 need to be activated, while the activation energy for P1 is usually higher than that of P2. Thus, the power of external stimulus (the EDR in our case) will decide which process prevails, and subsequently the direction and efficiency of the transition. Without external stimulus (e-beam blanked), the formed phase was found to be rather stable (Supplementary Fig. 9). In Fig. 5, an ultrahigh EDR 5.0 as well as an extended irradiation were used to push the F-to-C transition to its limit. Between Figs. 5g, h, continuous irradiation raised the gap in region #1 from -21 pm to -31 pm. In comparison, for region #2 with an initial gap of -25 pm, the e-beam was soon no longer able to attract more $V_O^{\cdot\cdot}$ to this region. Instead, the already formed C-type structure was activated to rotate, and finally a gap of -31 pm was detected in the perpendicular direction in Fig. 5h. This observation of C-type rotating instead of further forming could potentially account for the dropped slope in Fig. 3h for EDR 1.0. The highly dynamic diffusion of $V_O^{\cdot\cdot}$ causes not only the O lattice distortion, but also non-saturated chemical bonding. As plotted in Figs. 4c, d, the experimental $A_{O\_v}$ goes up to -0.04, corresponding roughly to a $\delta = 0.44$ and $M_{0.12}^{4+}M_{0.88}^{3+}O_{2-0.44}^{2-}$. For our sample, only 12% of the M sites are occupied by $Gd^{3+}$ (Supplementary Note 1 and Fig. 1), which is incapable of migration. Thus, within the irradiated region, considerable $Ce^{4+}$ has to be reduced to $Ce^{3+}$. As shown in Fig. 6c, the local Ce valences vary significantly, which could facilitate a substantial electron exchange between the $Ce^{4+}$ and $Ce^{3+}$ inside and outside the irradiated region.

Including all the observed features, a possible transition process is proposed in Fig. 7. Assuming an average composition of $Ce_{0.67}^{4+}Ce_{0.21}^{3+}Gd_{0.12}^{3+}O_{2-0.165}^{2-}$ with F-type structure (F-CGO33, 33% of the metal sites are occupied by $M^{3+}$), Fig. 7a depicts the early stage of an F-to-C transition along < 001 >. One unit-cell of the F-type is outlined at the upper-left corner. Two regions, labeled as $A_a^{\parallel}$ and $A_a^{\perp}$, adopt the C-type structure (C-CGO40). The superscript $\parallel$ and $\perp$ refer to the two perpendicular orientations of the C-type structure. Due to charge compensation, the substrate changes accordingly to F-CGO32 as indicated at the lower-left corner. The rising r in Fig. 3e suggests an expansion of the C-type under e-beam irradiation. Thus, in Fig. 7b, two

new regions, C-CGO50 and C-CGO40, are introduced as $B_b^{\parallel}$ and $B_b^{\perp}$. Besides, Figs. 4c, d suggest accumulating $V_O^{\cdot\cdot}$ and $Ce_M'$ under the e-beam irradiation. Therefore, the $A_a^{\parallel}$ and $A_a^{\perp}$ (both C-CGO40) evolve to $A_b^{\parallel}$ (C-CGO80) and $A_b^{\perp}$ (C-CGO60) respectively. As a result, the substrate is now F-CGO16. In Fig. 7c, the e-beam irradiation further pushes the $V_O^{\cdot\cdot}$ and $Ce_M'$ concentration, leading to $B_c^{\parallel}$ (C-CGO60), $B_c^{\perp}$ (C-CGO60), and $A_c^{\perp}$ (C-CGO80). Moreover, instead of achieving even higher $V_O^{\cdot\cdot}$ and $Ce_M'$ concentration, the $A_b^{\parallel}$ in Fig. 7b experiences a 90° rotation and integrates into $A_c^{\perp}$ in Fig. 7c, similar to the observation in Fig. 5b. All these changes then leave a substrate with F-CGO9. Table 1 lists the compositions for the different regions. Corresponding to the outlined rectangles in Figs. 7a–c, Figs. 7d–f list the simulated HRTEM images. The experimentally observed features, including the broader oscillation of $d_{O\_x}$ and 90° rotation of the C-type structure, are well embodied in the simulations.

In summary, the reversible phase transitions of CGO were explored in situ with picometer precision, employing a combination of advanced TEM techniques. With proper EDRs, the transitions can be accelerated, retarded, or reversed. The simultaneous visualization of both M and O sites allows a precise determination of lattice distortions associated with the transition, and further sheds light on the local $V_O^{\cdot\cdot}$ concentration. Besides, the C-type structure was observed orienting along two perpendicular yet competing directions, suggesting that the e-beam can not only alter the $V_O^{\cdot\cdot}$ concentration but also manipulate the C-type orientation. In the end, a collective rearrangement of $V_O^{\cdot\cdot}$ and $Ce_M'$ stimulated by e-beam was proposed to account for the observed transitions, which calls for further numeric calculation to establish a solid theory. Our findings demonstrate a largely controllable phase transition of ceria, unravel the associated $V_O^{\cdot\cdot}$ dynamics with unprecedented resolution and showcase great promise for diverse applications. Boosting the performance of ceria-based catalysts and electrolytes would be possible by remarkably reducing the operating temperature, as the redox of ceria is realized at room temperature in our case. Benefitting from the convenient design of ceria with desired properties through adjusting the $V_O^{\cdot\cdot}$ concentration, ceria-based memristors are to be advanced with tunable on/off ratios and superior memory density. The same principle could also be extended to other functional oxides.

## Methods

### Pellet sample preparation

Dense pellet with a nominal composition of 85 wt.% $Ce_{0.8}Gd_{0.2}O_{2-\delta}$:15 wt.% $FeCo_2O_4$ (85CGO20-FCO) was prepared by solid state reactive sintering (SSRS) method[58,59]. The phase ratio shown in the nominal composition refers to the initial weight ratio of CGO20 to FCO in the powder precursors without considering any phase interactions after sintering. To prepare the powder precursors, commercially available powders of CGO20 (Treibacher Industrie AG, Austria) as well as $Fe_2O_3$ and $Co_3O_4$ (Sigma-Aldrich, Germany) were homogenized by ball milling in ethanol. Details of the ball milling procedures can be found in ref.[58] After drying, the powder mixtures were pressed into pellets and sintered at 1200 °C for 10 h. During cooling, a slow rate of 0.5 °C/min was used between 900 °C and 800 °C in order to avoid cracking[60], while for the other temperature ranges, a faster cooling rate of 3 °C/min was used. Based on the previous work[32,33], the CGO grain interior in the composite is not affected by the FCO phase, and thus is chosen for this study.

## TEM sample preparation

A pallet of 85CGO20-FCO was first embedded in resin (Kulzer). After gradual grinding and polishing, a bulk cross-sectional sample with a flat surface was prepared. TEM cross-sectional specimens were cut from the polished bulk sample by focused ion beam (FIB) milling using an FEI Strata 400 system with a Ga ion beam. Carbon coating was applied to reduce the possible charging problem during FIB preparation. Further thinning and cleaning were performed with an Ar ion beam in a Fischione Nanomill 1040 using 900 and 500 eV beam energy in sequence.

## Characterization techniques

TEM bright field imaging and electron diffraction were performed by JEOL JEM F200 at 200 kV accelerating voltage. The image time series with various EDRs (in Fig. 3) were recorded in a consecutively way from the same region. The tilted sample was left inside the microscope for hours before imaging, to minimized potential sample drift. The diffraction patterns (in Fig. 2g) were recorded from a region with -120 nm in diameter. HRTEM NCSI was recorded with 300 kV accelerating voltage by Spectra 300, equipped with a high-brightness X-FEG monochromated source, a piezo-enhanced CompuStage, and two Cs correction optics. The CETCOR below the objective lens can be used for high-resolution TEM imaging with a bottom-mounted, retractable, fast Ceta CMOS camera. HRTEM image simulation was performed using Dr. Probe under the experimental condition with a typical sample thickness around 5 nm. The structural models were visualized with VESTA[61]. High resolution HAADF and iDPC imaging, EDX chemical mapping and EELS SI with atomic resolution were conducted with 200 kV accelerating voltage (unless otherwise noted) in an FEI Titan G2 80-200 ChemiSTEM microscope equipped with an XFEG, a probe Cs corrector, a super-X EDX system, and a Gatan Enfinium ER (model 977) spectrometer with DUAL EELS acquisition capability[62]. The convergence semi-angle for STEM imaging and EDX chemical mapping was approximately 22 mrad, while the collection semi-angles were 80-200 mrad for HAADF imaging, around 10-60 mrad for iDPC imaging, and around 47 mrad for EELS spectrum imaging. HAADF and iDPC image simulation was performed using Dr. Probe under the experimental condition with a typical sample thickness around 8 nm. EELS spectrum images were recorded with 0.5 eV per channel energy dispersion and 0.1 s dwell time for each pixel. Multivariate statistical analysis (MSA) was performed to reduce the noise of the EEL spectra with weighted principle-component analysis (PCA).

## Data availability

The data that support the findings of this study are available within the paper and supplementary information files. Source data are provided in this paper.

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

## Acknowledgements

This work has been supported by the Deutsche Forschungsgemeinschaft (Project Number 387282673). K.R. acknowledges support by the Bundesministerium für Bildung und Forschung (NEU-ROTEC, 16ME0399, and 16ME0398K). The authors thanks Dr. Hongchu Du and Penghan Lu (Forschungszentrum Jülich) for valuable discussion.

## Author contributions

K.R. designed and performed the TEM experiments and data analysis. L.J. and J.M. provide insightful discussion on the TEM results. K.R. and L.J. performed the simulation of TEM images. F.Z. synthesized the CGO under the supervision of S.B. and W.A.M. The manuscript was written by K.R. with contributions from all authors.

## Funding

## Competing interests

The authors declare no competing interests.
