## [Peer Review File · Nature Communications]

in situ observation of reversible phase transitions in Gd-doped ceria driven by electron beam irradiationREVIEWER COMMENTS

Reviewer #1 (Remarks to the Author):

This manuscript describes the transformation between the fluorite structure and bixbyite structure (a pseudofluorite structure with ordered oxygen vacancies) in gadolinium doped cerium oxide. These two structures were referred to as F- and C-type, respectively. Electron irradiation was used to control the transition between the two structures. The electron dose rate was found to affect the rate at which the transformation occurred, and some rather elegant analysis of atomic resolution images enabled quantification of the local oxygen vacancy concentration. This technique enabled the observation of complex dynamics as the fluorite structure converted to the bixbyite structure, which included nucleation and subsequent changes in the orientation of the C-type domains.

I think the data quality and much of the analysis are excellent and because the topic of oxygen vacancy dynamics is important, I think this paper will be of interest and use to other researchers. However, some of the interpretation and discussion seems slightly superficial. It would be helpful to see some discussion encompassing some of the recent theories in electron beam damage and the observed results. For instance, a deeper mechanistic discussion on why the C-type domains in Figure 5 rearranged with additional irradiation would be informative. As another example, several times the potential for "tuning" of the structure was mentioned but no practical pathways for actually accomplishing that were discussed or demonstrated. Accordingly, I would suggest revisions be made. Additional comments and questions are listed below.

Comments and Questions:

1. Some of the language with regards to controlling the microstructure needs to be revised as it gives the perception of greater control of the phase transformation than what was demonstrated. For example, on Line 23 and 24, "the concentration and ordering direction of VO are subject to fine tuning via electron beam". I agree that Figure 5 demonstrates that the orientation of the C-Ce2O3 domains changes with additional irradiation and while the process may not be stochastic, it also was not intentionally manipulated or directed by selection of the electron beam parameters.

2. Lines 98 and 99: "{110} spots are also detected (indicated by the rhombus), which are likely a result of multiple diffraction."

This should be clarified. This seems to suggest the C-Ce2O3 {220} reflections (written as {110} in the manuscript to follow the F-type symmetry) are kinematically forbidden and would therefore require double diffraction to be excited. I don't think that is accurate as the {220} reflections are kinematically allowed in the C-Ce2O3 structure.

3. Lines 131 to 134: "Moreover, the intensity ratio r_A and r_B between the two {010} and the two 132 corresponding {020} spots along the indicated A and B directions in Fig. 3b are also plotted in 133 Fig. 3f, to study the anisotropy of the phase transition. Both r_A and r_B are growing at a similar 134 pace at the beginning for all the transitions."

Based on the cubic symmetry of the C-Ce2O3 structure – for example a kinetic diffraction simulation of [001] C-Ce2O3 indicates that the r_A and r_B ratios would be equal – it is not clear to me why this r_A and r_B ratio would correspond to differences in the anisotropy of the phase transformation. Rather it seems this r_A and r_B ratio would be sensitive to other factors, such as mistilt of the specimen. Can any fundamental explanation or validation of this metric and how it reflects anisotropy of the transformation be provided?

4. Lines 212 and 213: "Besides, intense e-beam could remove the O atoms from the samples as well, and the threshold energy to displace O atoms in ceria is estimated as ~27 eV.[reference 49]"

Using thermodynamic values from a paper on molecular species to derive displacement energies in a crystalline material seems dubious as the displacement energy will be sensitive to structure and the local coordination around the atom in question. Can the rationale used to derive this number be

explained more? Furthermore, the discussion in this paragraph seems incomplete. What is the maximum energy that can be transferred to the sample through elastic scattering at each accelerating voltage? Because the transformation from CeO₂ to C-Ce₂O₃ happened at a range of accelerating voltages (60 to 300 kV) are we to infer that some other process, not elastic scattering, is driving the transformation?

5. Lines 222 to 225: "The diffusion of VO was reported to be anisotropic, preferring Ce instead of Gd as nearest neighbors, and along due to the lowest energy barrier. [reference 32,33,48] In line with these observations, the VO ·· in our case are ordering along <001>, driven by the interaction with incident electrons."

No definitive evidence was provided to support that vacancies are ordering along the <001> directions. This conclusion seems to be made by conflating the relative orientation of the static displacements of the Ce atoms with the direction along which the oxygen vacancies are ordering. Therefore, the resulting conclusion is flawed. Using a preferential diffusion direction reported in another paper to infer the crystallographic direction along which oxygen vacancies order is not an entirely sound assumption.

6. Jiang (e.g., DOI:10.1016/j.micron.2016.02.007 and DOI:10.1016/j.micron.2023.103482) has discussed that electric fields induced by the electron beam can affect atomic diffusion, for which the illumination geometry (i.e., TEM vs. STEM) is an important parameter. Because both STEM and TEM were used here to study these samples, were any significant differences in behavior between the modes of illumination observed? Particularly, can any comments be made on how TEM versus STEM can be used to tune the transformation and microstructure?

Reviewer #2 (Remarks to the Author):

In this manuscript, the authors have performed a detailed study of Gd doped ceria that was studied via HRTEM under various electron dose rates. These dose rates allowed oxygen atoms to be desorbed, changing the stoichiometry of the ceria and causing reordering of the lattice to transform the structure from the F to the C structure. In this process, the cubic fluorite structure with a high degree of symmetry transforms into a lower symmetry structure, which gives rise to additional reflections in the diffraction patterns and changes in the contrast observed in HRTEM images.

The work was very carefully done and has a very high degree of technical sophistication. The level of detail the authors are able to determine from the TEM images is at the state of the art. It has allowed them to piece together at the atomic scale, how the Ce and O ions move as a function of the degree of reduction. The inclusion of STEM SI to study the fine structure of the Ce M_{4,5} is impressive. I have no concerns about the technical content of this manuscript, which is truly excellent. My comments below come from the possible broader implications of this work and also I felt that some published work had been overlooked.

I am aware of the work done by the group of Thierry at Lyon (Bugnet et al., Nano Lett. 2017, 17, 12, 7652–7658) who have shown how ceria gets reduced when exposed to the electron beam, but that the transformation could be stopped when oxygen was present in the E-TEM. This group had commented on the mechanism by which the O becomes mobile when exposed to the e-beam. I think the authors of this work should consider the mechanistic implications of this previously published study. Additionally, there is some previous work (Li et al., Phys. Rev. B. 2011, 84, 180201) explaining the F- to C-type lattice transition that occurs in Gd-doped ceria due to oxygen vacancies in the ceria matrix. I feel this work should be cited here.

The understanding of ceria is of great importance due to their use in many applications. The C structure has been implicated in the past as being responsible for certain catalytic properties. My concern about the current study is that while the authors very elegantly describe the dynamics of how O removal occurs in ceria particles, can this be related to any application of ceria? The paper would be strengthened if the authors could relate their observations to what happens to ceria powders and their performance. Also, another question is whether the C structure be generated

ex-situ in a powder and then moved to the microscope for study, or is this a phenomenon that is only going to occur in the microscope column under e-beam irradiation.

Reviewer #3 (Remarks to the Author):

Reviewer #4 (Remarks to the Author):

The techniques of data collection and analysis are the state of the art and beyond. It is especially impressive that the finest details of the vacancy ordering are visible. There is, however, a question of what the authors try to achieve. If the main point of the paper is to demonstrate the techniques, then the authors did it at the best of the levels. However, the fact that Gd-doped ceria with 30 mol% of Gd can exist in both phases disordered fluorite or bixbyite with partial occupancies is well known. In fact, even 25mol% of Gd is sufficient to trigger oxygen vacancy ordering if the conditions are right. The fact that there is a structural phase transition somewhere between 20 and 25mol% of Gd is mentioned in Ref. 32. The works of A. K. Tyagi preceded this work by more than a decade. However, their data did not get to ICSD. The works of Marco Scavini and Christina Artini investigated the order-disorder transition in 25% and in 30mol% of Gd. (some of their work)

Artini, Cristina; Pani, Marcella; Lausi, Andrea; Masini, Roberto; Costa, Giorgio A.; High temperature structural study of Gd-doped ceria by synchrotron X-ray diffraction (673 K \leq T \leq 1073 K); ICSD collection code 251488
Scavini, M.; Coduri, M.; Allietta, M.; Brunelli, M.; Ferrero, C.; Probing complex disorder in Ce_{1-x}Gd_xO_{2-x/2} using the pair distribution function analysis; (Ce_{0.25}Gd_{0.75})O_{1.625}; 90 [K] ICSD collection code 184588

The order-disorder transition depends on temperature (upon heating even many bixbyite structures with partial occupancies undergo disorder into a fluorite lattice) and kinetics (vacancy mobility). These two oppose each other: to get the vacancies ordered, they must be mobile, on the other hand, heating to increase their mobility by cause disorder. If the authors aimed at showing the capabilities of their methods, they had done superb, and it may serve as an example to others. However, this has to be reflected in the text, because the fact of order-disorder of vacancies in CGO30 is not known.

Technical question:

E-beam causes two effects: local heating and local reduction (from Ce⁴⁺ to Ce³⁺). It would be good if the authors could specify which of the effects is dominant and why they think so.

Reviewer #5 (Remarks to the Author):

The paper presents an impressive TEM investigation on the impact of intense electron beam irradiation on the oxygen vacancies ordering in Gd doped cerium oxide. The paper is well written and presents interesting results. However, two important shortcomings should be corrected before being suitable for publication in nature communications. In the "Methods" section it is described only the preparation of the 85 wt.% Ce_{0.8}Gd_{0.2}O_{2-δ}:15 wt.% FeCo₂O₄ (85CGO20-FCO) sample, while, to my understanding, the result and discussion section reports measurements on Ce_{0.88}Gd_{0.12}O_{2-δ}. The author should describe the synthesis of the analysed sample(s). In addition, the authors should ensure that the experimental conditions described in the "Characterisation Techniques" section refer to the analysed samples. The second problem is that it is not clear in the paper how the findings here described could be useful in the broader scenario of doped ceria compounds both for the comprehension of their physical properties and for their applications. Only the last sentence of the conclusion section

briefly touches on this topic.

Some further comments:

At line 229 it is stated that "For our sample, only 12% of the M sites are occupied by Gd³⁺" This is the first time where the actual cationic composition of the investigated sample has been described. Given that the Methods section has to be rewritten, providing this fundamental information at the beginning of the Results section (and in the Abstract) would help the reader.

Line 111-112: the dosing rate is expressed in $e^{-1\text{\AA}^{-2}\text{s}^{-1}}$ or $e^{+1\text{\AA}^{-2}\text{s}^{-1}}$?

For direct comparison of the d_{V-O} values, a scale on the y axis should be added to Figure S3

Line 178: the d_{O_l} parameter has not been defined in the text

According to several figures the d_{M-V} value does not change on varying Gd percentage or delta.

However, the displacement of one of the two M position in respect to the fluorite special site is huge and varies on increasing Gd concentration, In fact, while in CeO₂ each cation has 12 identical M-M distances, in pure Gd six shorter and six longer M-M distances exist (delta_distance being as large as 0.5 Å). This is well apparent in the scratches of simulations (e.g. in Fig.1 and Fig.S3).

Doesn't it affect the d_{M-V} values?

At lines 199-200 it is stated that the "Ce³⁺ ratios (i suppose it is the $[\text{Ce}^{3+}]/[\text{Ce}^{4+}]$ or the $[\text{Ce}^{3+}]/([\text{Ce}^{3+}]+[\text{Ce}^{4+}])$ ratios) vary from 2% to 34% in three different points. Later on (lines 219-221) the authors suggest that no oxygen evolution should be present. Instead, the beam should induce a rearrangement of Ce³⁺ and oxygen vacancies. Does this imply that the oxygen vacancies concentration delta is larger than x/2 also in the pristine sample? In addition, referring to Figure 6 and Fig.S4a, did the authors take some images of the #1, #2 and #3 zones where the EELS spectra on the Ce-M4-M5 edges have been measured? Did they note significant differences among them?

Summarising: in my opinion the paper has to be published after major revision

RESPONSE TO REVIEWERS' COMMENTS

We thank the reviewers for their time and the constructive input. In response, we have made a number of changes to our manuscript. Our responses to the reviewers together with corresponding changes in text are described below. Please refer to the reviewed version for the labels of Lines and Figures.

Reviewer #1 (Remarks to the Author):

This manuscript describes the transformation between the fluorite structure and bixbyite structure (a pseudofluorite structure with ordered oxygen vacancies) in gadolinium doped cerium oxide. These two structures were referred to as F- and C-type, respectively. Electron irradiation was used to control the transition between the two structures. The electron dose rate was found to affect the rate at which the transformation occurred, and some rather elegant analysis of atomic resolution images enabled quantification of the local oxygen vacancy concentration. This technique enabled the observation of complex dynamics as the fluorite structure converted to the bixbyite structure, which included nucleation and subsequent changes in the orientation of the C-type domains.

I think the data quality and much of the analysis are excellent and because the topic of oxygen vacancy dynamics is important, I think this paper will be of interest and use to other researchers. However, some of the interpretation and discussion seems slightly superficial. It would be helpful to see some discussion encompassing some of the recent theories in electron beam damage and the observed results. For instance, a deeper mechanistic discussion on why the C-type domains in Figure 5 rearranged with additional irradiation would be informative. As another example, several times the potential for “tuning” of the structure was mentioned but no practical pathways for actually accomplishing that were discussed or demonstrated. Accordingly, I would suggest revisions be made. Additional comments and questions are listed below.

Response: we thank the reviewer for the suggestion. Accordingly, we have revised the discussion part starting from Line 202, and included an extra discussion on Fig. 5. Besides, the text regarding the potential for structural tuning has been revised. For details, please refer to the Responses to point 1, 4, and 6 below.

Comments and Questions:

1. Some of the language with regards to controlling the microstructure needs to be revised as it gives the perception of greater control of the phase transformation than what was demonstrated. For example, on Line 23 and 24, “the concentration and ordering direction of V_O are subject to fine tuning via electron beam”. I agree that Figure 5 demonstrates that the orientation of the C-Ce₂O₃ domains changes with additional irradiation and while the process may not be stochastic, it also was not intentionally manipulated or directed by selection of the electron beam parameters.

Response: we thank the reviewer for the suggestion. We have revised the sentence in Line 23 to “the V_O concentration and the orientation of the newly formed phase can be altered via electron beam”. In Line 143, the title “Fine tuning of the C-type” is changed to “Modifying the C-

type structure". In addition, the Introduction part starting from Line 58 and the Conclusion part have been modified accordingly.

After noticing the reversible F-to-C transition under TEM illumination, we carefully designed both experiment and data analysis to explore the process in detail. In addition to varying the EDRs to control the direction and efficiency of the transition, it's also interesting to know where is the limit of a F-to-C transition. As shown in Fig. 3, r_A and r_B split at the end of the recording, suggesting that all the formed C-type structure tends to orient along the same direction (either along a or b as noted in Fig. 1a). To check this, we further pushed the EDR to 5.0 ($\sim 2 \times 10^4 \text{ e} \cdot \text{\AA}^{-2} \cdot \text{s}^{-1}$) and prolonged the irradiation as in Fig. 5. Under this rather extreme condition, the consistent orientation in Fig. 5a was broken, as some of the C-type structure rotated by 90° in Fig. 5b. Indeed, the exact parameters to fine control such a rotation still need to be figured out, but additional flexibility of manipulating the formed C-type structure could already be expected. It should be mentioned, although the EDR 5.0 under NCSI condition is helpful to push the limit of the formed C-type, it is too strong for recording the F-to-C transition, as the F-type will immediately transfer to C-type. Further discussion regarding a possible scenario in Fig. 5 can be found in the Response to point 4.

2. Lines 98 and 99: "{110} spots are also detected (indicated by the rhombus), which are likely a result of multiple diffraction."

This should be clarified. This seems to suggest the C-Ce₂O₃ {220} reflections (written as {110} in the manuscript to follow the F-type symmetry) are kinematically forbidden and would therefore require double diffraction to be excited. I don't think that is accurate as the {220} reflections are kinematically allowed in the C-Ce₂O₃ structure.

Response: we agree that the {220} reflections in the C-type are kinematically allowed, and thank the reviewer for pointing it out. The sentence in Line 98 has been deleted, as well as the markers in Fig. 2g.

3. Lines 131 to 134: "Moreover, the intensity ratio r_A and r_B between the two {010} and the two 132 corresponding {020} spots along the indicated A and B directions in Fig. 3b are also plotted in 133 Fig. 3f, to study the anisotropy of the phase transition. Both r_A and r_B are growing at a similar 134 pace at the beginning for all the transitions."

Based on the cubic symmetry of the C-Ce₂O₃ structure – for example a kinetic diffraction simulation of [001] C-Ce₂O₃ indicates that the r_A and r_B ratios would be equal – it is not clear to me why this r_A and r_B ratio would correspond to differences in the anisotropy of the phase transformation. Rather it seems this r_A and r_B ratio would be sensitive to other factors, such as mistilt of the specimen. Can any fundamental explanation or validation of this metric and how it reflects anisotropy of the transformation be provided?

Response: we agree with the reviewer, that r_A and r_B should be equal based on the kinetic diffraction simulation, as in Fig. R1d using the model Ce_{0.125}Gd_{0.875}O_{1.562}¹ (ICSD: 184589, sample thickness ~ 10 nm). Following the F-type symmetry, all the {010} spots in Fig. R1d are nearly equal. For comparison, Fig. R1e is the dynamic simulation using the same model. Evidently, the (010) spot is much stronger than the (100) spots. Similarly, we constantly

observed the split of r_A and r_B from experiment. Thus, the electron-sample interaction in our case could be better interpreted as a dynamic process.

In addition, HRTEM image is simulated² as well in Fig. R1a, and Fig. R1b is the corresponding FFT pattern. Along the two defined directions (A and B) in Fig. R1b, two line profiles are plotted in Fig. R1c. Consistent with Fig. R1e, the $\{010\}$ peaks along A is much higher than those along B, while all the $\{020\}$ peaks are more or less comparable. Moreover, the stronger $\{010\}$ spots in reciprocal space are associated with the direction in real space, along which the M layer intensity and the d_o oscillate (in Fig. 1c). Therefore, the $\{010\}$ spots with different intensities in FFT, and consequently the split of r_A and r_B measured from experiment, are able to shed light on the orientation of the C-type.

As pointed out by the reviewer that mistilt of the specimen may result in different r_A and r_B . In Fig. 3f, the split of r_A and r_B is constantly detected at a later stage of all the F-to-C transitions, where all the transitions were stimulated in a consecutive way from the same sample region. Potential sample drift should be minimized, as the tilted sample was left inside the microscope for hours before the *in situ* experiment. Thus, if sample mistilt causes the split of r_A and r_B , it should be detectable from the beginning of all the transitions.

Accordingly, we have included the discussion above into the Supplementary Information, and revised the text in Line 135. Besides, in Line 294, experimental details regarding Fig. 3 has been provided.

Figure R1: (a-b) The simulated HRTEM image and the corresponding FFT pattern based on $\text{Ce}_{0.125}\text{Gd}_{0.875}\text{O}_{1.562}$ along $[001]$. A structural model is overlaid on the image. (c) The line profiles along the A and B as outlined in (b). (d-e) The simulated diffraction pattern from $\text{Ce}_{0.125}\text{Gd}_{0.875}\text{O}_{1.562}$ along $[001]$, based on kinematical and dynamical calculation respectively. All the indexing is following the F-type symmetry, and a sample thickness ~ 10 nm is used for simulation.

4. Lines 212 and 213: “Besides, intense e-beam could remove the O atoms from the samples as well, and the threshold energy to displace O atoms in ceria is estimated as ~ 27 eV.[reference 49]”

Using thermodynamic values from a paper on molecular species to derive displacement energies in a crystalline material seems dubious as the displacement energy will be sensitive to structure and the local coordination around the atom in question. Can the rational used to derive this number be explained more? Furthermore, the discussion in this paragraph seems incomplete. What is the maximum energy that can be transferred to the sample through elastic scattering at each accelerating voltage? Because the transformation from CeO₂ to C-Ce₂O₃ happened at a range of accelerating voltages (60 to 300 kV) are we to infer that some other process, not elastic scattering, is driving the transformation?

Response: we agree with the reviewer that the displacement energy is sensitive to the structure and the local coordination around the atom. A careful numeric calculation would be required to deduce a reliable displacement energy for the O inside ceria, which is however beyond the scope of the current study.

During the elastic scattering process, the amount of energy transferred to the nucleus, E , can be estimated as $E = E_{max} \sin^2(\theta/2)$, where $E_{max} \approx E_0(1 + E_0/1022keV)/(456A)$, E_0 is the kinetic energy of the primary electrons, and A is the atomic weight of the target atom. Different E_0 were used in our study (60, 200, and 300 keV), corresponding to $E_{max} \approx 8.7, 32.8, \text{ and } 53.2$ eV, respectively. Experimentally, F-to-C transition was observed for all the used E_0 . For an atom at a lattice site in a compact crystal, displacement usually requires ten of eV in most cases.³ Besides, as shown by the HRTEM time series in Supplementary Fig. S6, no significant loss of atoms can be detected with 300 keV. Thus, the energy transfer from elastic scattering process should only have limited contribution to the transitions observed in our case.

In addition to the knock-on and radiolysis mechanism, the theory of damage by induced electric field (DIEF) has successfully explained many phenomena caused by e-beam irradiation⁴⁻⁷. A related case to the current study is the phase transition between the perovskite structure and the O-deficient brownmillerite structure. Starting from a brownmillerite-structured SrFeO_{2.5} (BM-SFO), the built-up electric field during TEM imaging together with specimen heating (~200 to 300 °C) attracted adequate O²⁻ to the irradiated region. As a result, the BM-SFO was transferred to perovskite SrFeO₃ (P-SFO).⁸ This mechanism is further supported by *in situ* biasing experiment on La_{2/3}Sr_{1/3}MnO₃ (LSMO).⁹ An external electric field was applied (same direction as the field induced by TEM imaging according to the DIEF mechanism), and the BM-LSMO was transferred to P-LSMO by attracting the O²⁻ from the vicinity. It is worth mentioning, for the *in situ* biasing study, STEM imaging instead of TEM was chosen to exclude the contribution from TEM illumination.¹⁰ Applying the DIEF mechanism to our case, the built-up electric field by TEM illumination should attract O²⁻ to the irradiated region, and reduce the local $V_{\ddot{O}}$ concentration. This is however opposite to our observation, as the $V_{\ddot{O}}$ concentration is going up with continuous irradiation (Fig. 4c-d). Thus, although there are some similarities with the previous reports, our case can't be fully explained by the DIEF theory.

The exact mechanism which accounts for our observation should be complicated. Qualitatively, the phase transition consists of two competing processes: accumulating and ordering of the $V_{\ddot{O}}$ towards the irradiated region (P1, the forward F-to-C transition), and dissipating and disordering of the $V_{\ddot{O}}$ from the irradiated region (P2, the reversed F-to-C transition). Both P1 and P2 need to be activated, while the activation energy for P1 is usually higher than that of P2. Thus, the power of external stimuli (the EDR in our case) will decide which process prevails, and subsequently the direction and efficiency of the transition. With the e-beam blanked, the formed phase is found to be rather stable (Supplementary Fig. S7).

An extreme case is shown in Fig. 5, where the limit of a F-to-C transition was pushed with an ultrahigh EDR 5.0 and an extended irradiation. Between Fig. 5g and 5h, continuous irradiation raised the gap in region #1 from ~21 pm to ~31 pm by attracting more V_{O}^{\bullet} . In comparison, the e-beam was soon no longer able to attract more V_{O}^{\bullet} to region #2 with an initial gap of ~25 pm (slightly higher than the ~21 pm in region #1). Instead, the already formed C-type structure was activated to rotate, and finally a gap of ~31 pm was detected in the perpendicular direction in Fig. 5h. This observation of C-type rotating instead of further forming could potentially account for the dropped slope in Fig. 3h for EDR 1.0.

Nevertheless, to establish a solid theory, further numeric calculations are necessary, and our experimental observation would make a significant contribution to quantitatively clarify this matter.

Accordingly, we have deleted the citation of Ref. 49, and the discussion part starting from Line 202 has been revised.

5. Lines 222 to 225: “The diffusion of VO was reported to be anisotropic, preferring Ce instead of Gd as nearest neighbors, and along due to the lowest energy barrier. [reference 32,33,48] In line with these observations, the VO^{\bullet} in our case are ordering along $\langle 001 \rangle$, driven by the interaction with incident electrons.”

No definitive evidence was provided to support that vacancies are ordering along the $\langle 001 \rangle$ directions. This conclusion seems to be made by conflating the relative orientation of the static displacements of the Ce atoms with the direction along which the oxygen vacancies are ordering. Therefore, the resulting conclusion is flawed. Using a preferential diffusion direction reported in another paper to infer the crystallographic direction along which oxygen vacancies order is not an entirely sound assumption.

Response: we regret for the misinterpretation. Indeed, from the experiment, we can only measure the static displacements of the O atomic positions (the oscillation of d_{O}). This oscillation is a result of the splitting at O position viewed along $\langle 001 \rangle$, partially due to the ordering of V_{O}^{\bullet} according to the proposed C-type models.¹¹ Nevertheless, as the reviewer pointed out, the measured d_{O} modulation could only provide circumstantial evidence about the V_{O}^{\bullet} ordering. Besides, the favored diffusion direction in¹² is not necessarily equal to the V_{O}^{\bullet} ordering direction.

To avoid the misleading, we have deleted the sentences starting from Line 222 to Line 225, “The diffusion of V_{O}^{\bullet} was..., driven by the interaction with incident electrons.” Besides, the description of “direction of V_{O}^{\bullet} ordering” in the text has been revised as “C-type structure orientation”.

6. Jiang (e.g., DOI:10.1016/j.micron.2016.02.007 and DOI:10.1016/j.micron.2023.103482) has discussed that electric fields induced by the electron beam can affect atomic diffusion, for which the illumination geometry (i.e., TEM vs. STEM) is an important parameter. Because both STEM and TEM were used here to study these samples, were any significant differences in behavior between the modes of illumination observed? Particularly, can any comments be made on how TEM versus STEM can be used to tune the transformation and microstructure?

Response: we thank the reviewer for sharing the two reports, which provide valuable insights into the irradiation effects observed by TEM. We agree that the irradiation effects are strongly dependent on the illumination geometry, as we observed so. According to the DIEF theory,^{4,6} for the TEM mode, the resulted electric field is mainly along the beam direction at the irradiated surface, while has its maximum strength on the periphery of the irradiated region pointing outward. For the STEM mode, the induced electric fields are then perpendicular to the beam direction and pointing outwards. We have seen many experimental observations successfully explained by this mechanism.¹³⁻¹⁵ As in the Response to point 4, this theory should have limited contributions to our observation.

In our study, all the reversible F-to-C transitions were stimulated under TEM imaging condition. By varying the EDRs, both direction and efficiency of the transition, as well as the C-type orientation can be manipulated. In contrast, no transition has been observed in STEM mode, either with 200 keV or 300 keV (Supplementary Note 1 and Fig. S1).

The F-to-C transition can be considered as a collective movement of a large number of $V_{\text{O}}^{\bullet\bullet}$ s paired valence change of Ce, and these $V_{\text{O}}^{\bullet\bullet}$ s need to be simultaneously activated and rearranged. Considering the illumination geometry, the parallel beam in TEM mode (hundreds of nms in size) is able to interact with a large region of the sample at the same time. In contrast, a highly focused probe ($< 1 \text{ \AA}$) is used in STEM mode. During the scanning, only a small portion of the sample (the sample thickness is normally $\sim 10 \text{ nm}$ in our case) is directly interacting with the incident electrons each time. It is then unlikely that enough $V_{\text{O}}^{\bullet\bullet}$ s will be simultaneously stimulated by the probe, to induce a detectable phase transition.

Accordingly, the discussion part starting from Line 202 and the Supplementary Note 1 have been revised.

Reviewer #2 (Remarks to the Author):

In this manuscript, the authors have performed a detailed study of Gd doped ceria that was studied via HRTEM under various electron dose rates. These dose rates allowed oxygen atoms to be desorbed, changing the stoichiometry of the ceria and causing reordering of the lattice to transform the structure from the F to the C structure. In this process, the cubic fluorite structure with a high degree of symmetry transforms into a lower symmetry structure, which gives rise to additional reflections in the diffraction patterns and changes in the contrast observed in HRTEM images.

The work was very carefully done and has a very high degree of technical sophistication. The level of detail the authors are able to determine from the TEM images is at the state of the art. It has allowed them to piece together at the atomic scale, how the Ce and O ions move as a function of the degree of reduction. The inclusion of STEM SI to study the fine structure of the Ce M_{4,5} is impressive. I have no concerns about the technical content of this manuscript, which is truly excellent. My comments below come from the possible broader implications of this work and also I felt that some published work had been overlooked.

I am aware of the work done by the group of Thierry at Lyon (Bugnet et al., Nano Lett. 2017, 17, 12, 7652–7658) who have shown how ceria gets reduced when exposed to the electron beam, but that the transformation could be stopped when oxygen was present in the E-TEM.

This group had commented on the mechanism by which the O becomes mobile when exposed to the e-beam. I think the authors of this work should consider the mechanistic implications of this previously published study. Additionally, there is some previous work (Li et al., Phys. Rev. B. 2011, 84 ,180201) explaining the F- to C-type lattice transition that occurs in Gd-doped ceria due to oxygen vacancies in the ceria matrix. I feel this work should be cited here.

Response: we thank the reviewer for the suggestion. The report by Bugnet et al.¹⁶ provided direct observation regarding the surface mobility of ceria nanoparticles under different atmospheres by *in situ* ETEM. Substantial mobility of Ce atoms at the surface was noticed in high vacuum. Under O₂ and CO₂, either O-termination or carbonates-termination surface were determined, which significantly suppress the surface mobility. Li et al.¹⁷ pointed out that the F-to-C transition occurs at the atomic level through the defect cluster growth. A unique isosceles triangle structure, and subsequently a dumbbell structure acts as the building block that dominates the structure evolution from F to C type. **The two reports looked into the ceria redox process from complementary aspects, and we have cited them in our work.**

The understanding of ceria is of great importance due to their use in many applications. The C structure has been implicated in the past as being responsible for certain catalytic properties. My concern about the current study is that while the authors very elegantly describe the dynamics of how O removal occurs in ceria particles, can this be related to any application of ceria? The paper would be strengthened if the authors could relate their observations to what happens to ceria powders and their performance. Also, another question is whether the C structure be generated ex-situ in a powder and then moved to the microscope for study, or is this a phenomenon that is only going to occur in the microscope column under e-beam irradiation.

Response: we thank the reviewer for the suggestion. As mentioned by the reviewer, ceria is of great importance due to its promising application in the catalyst and solid electrolytes. Since its function is mainly based on the formation and mobility of the $V_{\text{O}}^{\bullet\bullet}$ s, it is therefore critical to gain details regarding the $V_{\text{O}}^{\bullet\bullet}$ s dynamics especially at the atomic scale.

As an active support for noble metals in catalysis, the performance of ceria relies on an efficient formation of $V_{\text{O}}^{\bullet\bullet}$.¹⁸ An O deficient CeO_{2- δ} layer within a CeO_{2- δ} @CeO₂ core-shell heterostructure was found to be responsible for a fast ionic conduction, resulting in a remarkable power output.¹⁹ In both cases, the local concentration of $V_{\text{O}}^{\bullet\bullet}$ (δ) is the key parameter to evaluate the ceria's performance. By correlating A_{O_v} and δ , our method offers a reliable estimation of δ , and thus a direct assessment of the ceria's property simply by imaging. In addition to capture the $V_{\text{O}}^{\bullet\bullet}$ dynamics with unprecedented temporal and spatial resolution, we also demonstrated that the $V_{\text{O}}^{\bullet\bullet}$ formation can be controlled via a proper stimulus (ERD in our case). As a result, flexibly designing the ceria with desired properties is possible. This is significantly important for its application in memristors,²⁰ as more than two states may be introduced, and a great boosting of the memristor's performance is to be expected. Another aspect would be the functioning temperature of ceria. Usually, the redox reaction of ceria takes place at high temperature (>600 K) and/or low oxygen partial pressure. In our case, the reversible F-to-C transition is realized at room temperature. Thus, our results showcase the potential of reducing operation temperature of both catalysts and solid oxide fuel cells.

Regarding the C-type structure observed in our case, we believe it is formed due to the TEM illumination, as our *in situ* experiment shows clearly F-to-C transitions. Besides, in STEM

mode, where the illumination geometry is unlikely to stimulate the phase transition,⁹ we always detected pure F-type structure.

Accordingly, additional comments on our results regarding their contribution to the ceria-based application have been included in the Conclusion part.

Reviewer #3 (Remarks to the Author):

Response: We thank the reviewer for his/her time and efforts, and hope that all the raised concerns have been addressed.

Reviewer #4 (Remarks to the Author):

The techniques of data collection and analysis are the state of the art and beyond. It is especially impressive that the finest details of the vacancy ordering are visible.

There is, however, a question of what the authors try to achieve. If the main point of the paper is to demonstrate the techniques, then the authors did it at the best of the levels. However, the fact that Gd-doped ceria with 30 mol% of Gd can exist in both phases disordered fluorite or bixbyite with partial occupancies is well known. In fact, even 25mol% of Gd is sufficient to trigger oxygen vacancy ordering if the conditions are right. The fact that there is a structural phase transition somewhere between 20 and 25mol% of Gd is mentioned in Ref. 32. The works of A. K. Tyagi preceded this work by more than a decade. However, their data did not get to ICSD. The works Marco Scavini and Christina Artini investigated the order-disorder transition in 25% and in 30mol% of Gd. (some of their work)

Artini, Cristina; Pani, Marcella; Lausi, Andrea; Masini, Roberto; Costa, Giorgio A.; High temperature structural study of Gd-doped ceria by synchrotron X-ray diffraction (673 K ≤ T ≤ 1073 K); ICSD collection code 251488

Scavini, M.; Coduri, M.; Allieta, M.; Brunelli, M.; Ferrero, C.; Probing complex disorder in Ce_{1-x}Gd_xO_{2-x/2} using the pair distribution function analysis; (Ce_{0.25}Gd_{0.75})O_{1.625}; 90 [K] ICSD collection code 184588

Response: we thank the reviewer for sharing the literature. Indeed, the ceria doped by different amount of Gd and the associated ordering-disordering have been studied for a long time. All the work mentioned above represent important contributions to understand the system. Therefore, we have cited them^{1,21,22} in our work.

The order-disorder transition depends on temperature (upon heating even many bixbyite structures with partial occupancies undergo disorder into a fluorite lattice) and kinetics (vacancy mobility). These two oppose each other: to get the vacancies ordered, they must be mobile, on the other hand, heating to increase their mobility by cause disorder.

Response: we agree with the reviewer that the phase transition is a competition between the ordering and disordering process. Based on the calculation in the Supplementary Note 5 and Table 1, heating effect induced by the TEM illumination is rather ignorable. As discussed above, the exact mechanism of the phase transition is complicated. Qualitatively, it is reasonable to expect, that both ordering and disordering process need to be activated, and the activation energy is higher for the ordering process than that for the disordering. Depending on how much energy can be received by the sample (sensitive to the applied EDR in our case), either a forward F-to-C (ordering) or a reversed F-to-C (disordering) transition could be stimulated.

If the authors aimed at showing the capabilities of their methods, they had done superb, and it may serve as an example to others. However, this has to be reflected in the text, because the fact of order-disorder of vacancies in CGO30 is not known.

Response: we thank the reviewer for the suggestion. Accordingly, additional highlight of our method has been included in the Introduction and Conclusion part.

Technical question:

E-beam causes two effects: local heating and local reduction (from Ce⁴⁺ to Ce³⁺). It would be good if the authors could specify which of the effects is dominant and why they think so.

Response: we thank the reviewer for pointing it out. As in Supplementary Note 5 and Table 1, the heating effect in our case is rather limited. During a F-to-C transition, the local V_{O} concentration within the irradiated region is going up. To cope with it, the Ce within the same region will also be largely reduced to Ce³⁺. As determined by the EELS in Fig. 6, the Ce valence in our sample is a mixture of Ce³⁺ and Ce⁴⁺, and electron hopping between neighboring Ce³⁺ and Ce⁴⁺ would be relatively easy. Therefore, our observation could be described as a collective rearrangement of V_{O} 's paired with valence change of Ce. Combining the suggestions from other reviewers, we have revised the discussion part starting from Line 202.

Reviewer #5 (Remarks to the Author):

The paper presents an impressive TEM investigation on the impact of intense electron beam irradiation on the oxygen vacancies ordering in Gd doped cerium oxide.

The paper is well written and presents interesting results. However, two important shortcomings should be corrected before being suitable for publication in nature communications.

In the “Methods” section it is described only the preparation of the 85 wt.% Ce_{0.8}Gd_{0.2}O_{2-δ}:15 wt.% FeCo₂O₄ (85CGO20-FCO) sample, while, to my understanding, the result and discussion section reports measurements on Ce_{0.88}Gd_{0.12}O_{2-δ}. The author should describe the synthesis of the analysed sample(s). In addition, the authors should ensure that the experimental conditions described in the “Characterisation Techniques” section refer to the analysed samples.

Response: we regret for the misleading. A dual phase ceramic, 85 wt.% $\text{Ce}_{0.8}\text{Gd}_{0.2}\text{O}_{2-\delta}$:15 wt.% FeCo_2O_4 (85CGO20-FCO), is used in our work. Based on our previous work,²³ the CGO phase is not affected by the existence of the FCO phase, as only slight segregation of Fe and Co along the CGO grain boundary is noticed. Within the CGO grain interior, signal from either Fe or Co is not detectable. Thus, we focused on the CGO phase (calibrated as $\text{Ce}_{0.88}\text{Gd}_{0.12}\text{O}_{2-\delta}$ in Supplementary Note 4 and Fig. S8), to study the reversible phase transition.

We thank the reviewer for pointing it out. The Characterization techniques part has been revised accordingly.

The second problem is that it is not clear in the paper how the findings here described could be useful in the broader scenario of doped ceria compounds both for the comprehension of their physical properties and for their applications. Only the last sentence of the conclusion section briefly touches on this topic.

Response: we thank the reviewer for pointing it out. Combining the suggestions from other reviewers, extra comments on our results regarding their contribution to the ceria-based application have been included in the Conclusion part.

Some furthers comments:

At line 229 it is stated that “For our sample, only 12% of the M sites are occupied by Gd3+” This is the first time where the actual cationic composition of the investigated sample has been described. Given that the Methods section has to be rewritten, providing this fundamental information at the beginning of the Results section (and in the Abstract) would help the reader.

Response: we thank the reviewer for pointing it out. The sample information has been given in Line 19 (in Abstract), “Taking the Gd-doped ceria ($\text{Ce}_{0.88}\text{Gd}_{0.12}\text{O}_{2-\delta}$) as a...”, and in Line 58 (in Introduction), “...is chosen for our study ($\text{Ce}_{0.88}\text{Gd}_{0.12}\text{O}_{2-\delta}$, in Supplementary Fig. S1 and Note 1)”.

Line 111-112: the dosing rate is expressed in $\text{e}^{-1}\text{Å}^{-2}\text{s}^{-1}$ or $\text{e}^{+1}\text{Å}^{-2}\text{s}^{-1}$?

Response: we thank the reviewer for pointing it out. It should be $\text{e}\cdot\text{Å}^{-2}\cdot\text{s}^{-1}$, and has been corrected in the text.

For direct comparison of the d_{V-O} values, a scale on the y axis should be added to Figure S3

Response: we thank the reviewer for pointing it out. Fig. S3 has been revised.

Line 178: the dO_I parameter has not been defined in the text

Response: we thank the reviewer for pointing it out. In Line 178, it has been changed to “the vertically averaged $d_{O,L}$ (the distance between neighboring O position along the lateral direction) as plotted...”.

According to several figures the d_{M-V} value does not change on varying Gd percentage or delta. However, the displacement of one of the two M position in respect to the fluorite special site is huge and varies on increasing Gd concentration, In fact, while in CeO₂ each cation has 12 identical M-M distances, in pure Gd six shorter and six longer M-M distances exist (delta_distance being as large as 0.5 Å). This is well apparent in the scratches of simulations (e.g. in Fig.1 and Fig.S3). Doesn't it affect the d_{M-V} values?

Response: we agree with the reviewer that part of the M sites are splitting when viewing along <001>. As a result, elongated shapes at these M sites show up in the simulated HRTEM images (in Fig. 1c and Fig. S3). Also, the splitting varies on increasing Gd concentration.

To measure the M-M or O-O distance in our case, each atomic position needs to be located first, by fitting with a 2D-Gaussian.²⁴ One example is shown in Fig. R2a-b. Fig. R2a is adopted from Fig. R1a, with part of the Ce_{0.125}Gd_{0.875}O_{1.562} model overlaid. The Fig. R2a is further duplicated in Fig. R2b, where the orange and green dots represent the determined M and O positions using the 2D-Gaussian method. As shown in Fig. R2b, regardless of the elongation, centers of those elongated M positions are always located. Based on the 2D-Gaussian fitting, d_1 - d_4 as well as the ellipticity at each atomic position are measured from Fig. R2c (a duplicate of Fig. R1a) and compared in Fig. R2d-e. In Fig. R2d, along the vertical direction, the modulation of both d_2 and ellipticity at the M sites are significant, while d_1 is rather constant. In Fig. R2e, along the lateral direction, only slight oscillations can be noticed for d_3 and ellipticity at the O sites. Thus, the M-M distance is not affected by the split at M positions in spite of the manifest elongation (ellipticity up to 1.32 is estimated from Fig. R2c). Besides, we noticed that it is rather challenging to resolve the elongation at M sites by experiment, probably due to imperfect imaging condition (noise, instrument/sample instability, residual astigmatism et al.). Overall, we haven't noticed any significant modulation of d_M from the experiment.

Fig. R2 (a) HRTEM image adopted from Fig. R1a with part of the $\text{Ce}_{0.125}\text{Gd}_{0.875}\text{O}_{1.562}$ model overlaid. (b) The same as (a), with the located atomic positions by 2D-Gaussian marked. The orange and green dots are for M and O sites, respectively. (c) A duplicate of Fig. R1a. (d) Laterally averaged d1, d2, and the estimated ellipticities at each M and O sites. (e) Vertically averaged d3, d4, and the estimated ellipticities at the M and O sites.

At lines 199-200 is stated that the “Ce³⁺ ratios (i suppose it is the $[\text{Ce}^{3+}]/[\text{Ce}^{4+}]$ or the $[\text{Ce}^{3+}]/([\text{Ce}^{3+}]+[\text{Ce}^{4+}])$ ratios) vary from 2% to 34% in three different points. Later on (lines 219-221) the authors suggest that no oxygen evolution should be present. Instead, the beam should induce a rearrangement of Ce³⁺ and oxygen vacancies. Does this imply that the oxygen vacancies concentration δ is larger than $x/2$ also in the pristine sample? In addition, referring to Figure 6 and Fig.S4a, did the authors take some images of the #1, #2 and #3 zones where the EELS spectra on the Ce-M4-M5 edges have been measured? Did they note significant differences among them?

Response: we regret for the misleading and thank the reviewer for pointing it out. The Ce³⁺ ratio means $\text{Ce}^{3+}/(\text{Ce}^{3+}+\text{Ce}^{4+})$, and corresponding changes have been made in the text in Line 199. In Line 219-221, we intended to suggest that in our case, the sample is not significantly exchanging oxygen with the outside environment during the phase transitions. Instead, the already existed V_{O} s inside the sample are rearranging. During the F-to-C transition, the irradiated region experienced an increase of Ce³⁺ and V_{O} , by attracting them from the non-irradiated region. Overall, the V_{O} content inside the sample is expected to be stable. Combining the suggestions from other reviewers, the discussion part starting from Line 202 has been revised.

After the EELS SI, we didn't notice any significant structural change. Fig. R3b is an overview HAADF image recorded between EELS measurement for region #2 and #3, as outlined by the rectangles. No significant structural changes can be noticed. Fig. R3c-d are the simultaneously recorded HAADF and iDPC images after all the EELS measurements and

around the scanned regions. To resolve the O sites, a rather high magnification was used, which inevitably reduces the field of view. Nevertheless, based on the FFTs (upper right insets in Fig. R3c-d), only F-type structure can be detected. Thus, the STEM illumination condition has rather limited contribution to cause detectable structural change in our case.⁹ Accordingly, **Supplementary Fig. S4 has been replaced by Fig. R3.**

Fig. R3 (a) HAADF image shows two CGO grains (A and B). Three locations are marked, corresponding to the three EELS curves in Figure 6c. (b) HAADF image recorded between collecting EELS signal from location #2 and #3. The scanning regions for #2 and #3 are outlined by the rectangles. (c-d) The simultaneously acquired HAADF and iDPC images recorded around the three locations after the EELS measurement. No extra spots can be noticed in the FFT patterns (upper right insets), suggesting no significant structural changes caused by STEM imaging or EELS measurements.

Summarising: in my opinion the paper has to be published after major revision.

Response: We thank the reviewer for his/her time and efforts, and hope that all the raised concerns have been addressed.

References

- 1 Scavini, M., Coduri, M., Allieta, M., Brune, M. & Ferrero, C. Probing Complex Disorder in $Ce_{1-x}Gd_xO_{2-x/2}$ Using the Pair Distribution Function Analysis. *Chem Mater* **24**, 1338-1345 (2012).
- 2 Barthel, J. Dr. Probe: A software for high-resolution STEM image simulation. *Ultramicroscopy* **193**, 1-11 (2018).
- 3 Egerton, R. F. Radiation damage to organic and inorganic specimens in the TEM. *Micron* **119**, 72-87 (2019).
- 4 Jiang, N. Electron irradiation effects in transmission electron microscopy: Random displacements and collective migrations. *Micron* **171**, 103482 (2023).
- 5 Jiang, N. Beam damage by the induced electric field in transmission electron microscopy. *Micron* **83**, 79-92 (2016).
- 6 Jiang, N. Electron beam damage in oxides: a review. *Rep Prog Phys* **79**, 016501 (2016).
- 7 Tao, J. *et al.* Reversible structure manipulation by tuning carrier concentration in metastable Cu_2S . *P Natl Acad Sci USA* **114**, 9832-9837 (2017).
- 8 Yang, Z. Z. *et al.* Guided anisotropic oxygen transport in vacancy ordered oxides. *Nat Commun* **14**, 6068 (2023).
- 9 Yao, L. D., Inkinen, S. & van Dijken, S. Direct observation of oxygen vacancy-driven structural and resistive phase transitions in $La_{2/3}Sr_{1/3}MnO_3$. *Nat Commun* **8** (2017).
- 10 Yao, L. D. *et al.* Electron-Beam-Induced Perovskite-Brownmillerite-Perovskite Structural Phase Transitions in Epitaxial $La_{2/3}Sr_{1/3}MnO_3$ Films. *Adv Mater* **26**, 2789-2793 (2014).
- 11 Scavini, M. *et al.* Percolating hierarchical defect structures drive phase transformation in $Ce_{1-x}Gd_xO_{2-x/2}$: a total scattering study. *Iucrj* **2**, 511-522 (2015).
- 12 Zhu, L. *et al.* Visualizing Anisotropic Oxygen Diffusion in Ceria under Activated Conditions. *Phys Rev Lett* **124**, 056002 (2020).
- 13 Ahluwalia, R. *et al.* Manipulating Ferroelectric Domains in Nanostructures Under Electron Beams. *Phys Rev Lett* **111** (2013).
- 14 Ma, J. Y. *et al.* Real-time observation of phase coexistence and a1/a2 to flux-closure domain transformation in ferroelectric films. *Acta Mater* **193**, 311-317 (2020).
- 15 Shin, D. *et al.* Preferential hole defect formation in monolayer WSe_2 by electron-beam irradiation. *Phys Rev Mater* **5** (2021).
- 16 Bugnet, M., Overbury, S. H., Wu, Z. L. & Epicier, T. Direct Visualization and Control of Atomic Mobility at {100} Surfaces of Ceria in the Environmental Transmission Electron Microscope. *Nano Lett* **17**, 7652-7658 (2017).
- 17 Li, Z. P. *et al.* Structural phase transformation through defect cluster growth in Gd-doped ceria. *Phys Rev B* **84** (2011).
- 18 Esch, F. *et al.* Electron localization determines defect formation on ceria substrates. *Science* **309**, 752-755 (2005).
- 19 Wang, B. Y. *et al.* Fast ionic conduction in semiconductor CeO_{2-6} electrolyte fuel cells. *Npg Asia Mater* **11** (2019).
- 20 Gao, P. *et al.* Electrically Driven Redox Process in Cerium Oxides. *J Am Chem Soc* **132**, 4197-4201 (2010).
- 21 Artini, C., Pani, M., Lausi, A., Masini, R. & Costa, G. A. High Temperature Structural Study of Gd-Doped Ceria by Synchrotron X-ray Diffraction ($673\text{ K} \leq T \leq 1073\text{ K}$). *Inorg Chem* **53**, 10140-10149 (2014).
- 22 Grover, V., Achary, S. N. & Tyagi, A. K. Structural analysis of excess-anion C-type rare earth oxide:: a case study with $Gd_{1-x}Ce_xO_{1.5+x/2}$ ($x= 0.20$ and 0.40). *J Appl Crystallogr* **36**, 1082-1084 (2003).
- 23 Ran, K. *et al.* Tuning the ceria interfaces inside the dual phase oxygen transport membranes. *Acta Mater* **226**, 117603 (2022).
- 24 Nord, M., Vullum, P. E., MacLaren, I., Tybell, T. & Holmestad, R. Atomap: a new software tool for the automated analysis of atomic resolution images using two-dimensional Gaussian fitting. *Advanced Structural and Chemical Imaging* **3**, 1-12 (2017).

REVIEWER COMMENTS

Reviewer #1 (Remarks to the Author):

I think the authors have satisfactorily addressed the reviewers' concerns. I have no additional comments. The paper is suitable for editing and publication.

Reviewer #2 (Remarks to the Author):

The authors have diligently addressed the comments by all of the reviewers. I am satisfied that the work is of high quality and scientifically interesting. In their concluding statements, they point out the broader impacts of their work in terms of the ability to modify the oxygen vacancy concentrations and the order-disorder in the ceria lattice at room temperature. These factors are all known to affect the properties of ceria. My question, yet unanswered in the revised manuscript, is whether these changes in ceria can be generated via any other means, such as thermal treatments in reducing atmospheres, and also if the electron beam irradiation is the only way to achieve these structures, can the sample be exposed to air and will it preserve the structures generated in the electron microscope. This will help guide future researchers in how to make use of these scientifically interesting findings.

The work is ready for publication and I do not need to see it again.

Reviewer #4 (Remarks to the Author):

The manuscript has improved considerably. However, there is one critical point missing: an explanation of how the e-beam causes the phase transition. This is the main novelty of this work, as the existence of ordering to the C-phase, promoted by reduction and cooling, is already known from previous studies.

The authors claim that the main novelty is the ability to control the C-F transition using the e-beam. This is a strong reason for publication in a leading journal, but since it is the key point, it needs to be clearly explained.

Reviewer #5 (Remarks to the Author):

The authors suitably replied to all my concerns.
I suggest publication

RESPONSE TO REVIEWERS' COMMENTS

Reviewer #4 (Remarks to the Author):

The manuscript has improved considerably. However, there is one critical point missing: an explanation of how the e-beam causes the phase transition. This is the main novelty of this work, as the existence of ordering to the C-phase, promoted by reduction and cooling, is already known from previous studies.

The authors claim that the main novelty is the ability to control the C-F transition using the e-beam. This is a strong reason for publication in a leading journal, but since it is the key point, it needs to be clearly explained.

Response: we thank the reviewer for the compliment, and agree that it's important to understand how the e-beam causes the phase transition. In this work, we are trying to demonstrate that combining advanced TEM techniques, the phase transition of Gd-doped ceria can be stimulated, as well as be in situ captured with ultrahigh precision. Our work explores the possibility to modulate the transition process via e-beam, unveils the oxygen dynamics at pm scale with sufficient temporal resolution, and also enables a reliable determination of the local V_O^{\bullet} concentration based on NCSI imaging. Supported by image simulation and EELS analysis, a collective rearrangement of V_O^{\bullet} paired with valence change of Ce was proposed to qualitatively describe the observed phase transition, which was also suggested by the reviewer previously (a competition between ordering and disordering of V_O^{\bullet} where the mobility of V_O^{\bullet} is a key factor).

Regarding the switching mechanism, several possibilities were discussed in the manuscript. Since no heating (the e-beam induced heating is nearly ignorable) or reduction environment has been applied during our experiment, the phase transition must be stimulated by the e-beam irradiation. Inside TEM, the interaction between the incident electrons and the sample is in general a rather complicated process, which could lead to knock-on damage, ionization damage and electrostatic charging.¹ However, as described in the section "The migration of V_O^{\bullet} and Ce_M' ", our observations can't be fully explained by these established mechanisms (no significant loss of atoms can be detected even at 300 kV, the ionization would be irreversible, and the e-beam induced field should cause opposite phenomena).

Thus, to deduce a quantitative explanation of the phase transition, a dedicated numerical calculation is required, and the following points learnt from the experiment should be considered: a) the F-to-C transition needs high EDRs, and the higher the EDRs, the faster the F-to-C transition; b) the C-to-F transition needs much lower EDRs, while no transition takes place with zero EDR; c) the phase transition can only be observed in TEM mode (parallel beam with 100s nm in size) but not in STEM mode (focused beam with probe size $<1 \text{ \AA}$). However, this would be beyond the scope of this work, but could be an interesting topic for a follow-up study.

References

- 1 Egerton, R. F. Radiation damage to organic and inorganic specimens in the TEM. *Micron* **119**, 72-87 (2019).

REVIEWERS' COMMENTS

Reviewer #4 (Remarks to the Author):

The work is fine. the experimental skills are spectacular. The authors claim that the transition is not triggered by heating, by reduction or by beam damage. Then by what?

The authors should either suggest something plausible or state in the main text explicitly that the trigger for the phase transition is unknown. There is nothing wrong with admitting that for now it is so. It does not chip away the value of the work.

RESPONSE TO REVIEWERS' COMMENTS

Reviewer #4 (Remarks to the Author):

The work is fine. the experimental skills are spectacular. The authors claim that the transition is not triggered by heating, by reduction or by beam damage. Then by what?

The authors should either suggest something plausible or state in the main text explicitly that the trigger for the phase transition is unknown. There is nothing wrong with admitting that for now it is so. It does not chip away the value of the work.

Response: we thank the reviewer for the suggestion. Indeed, after a detailed discussion considering all the established theories, we couldn't find one to fully explain our observations. Therefore, we proposed a follow-up numerical calculation in the conclusion part to clarify the matter quantitatively, which is however beyond the scope of this study.

Accordingly, the first sentence in the last paragraph in Page 9, "The exact mechanism which accounts for the phase transition should be complicated." is changed to "The exact mechanism which accounts for the phase transition is still to be defined."